# From Concepts to Components: Concept-Agnostic Attention Module Discovery in Transformers

**Jingtong Su**[*]
NYU & Meta AI, FAIR

**Julia Kempe**[†]
NYU & Meta AI, FAIR

**Karen Ullrich**[†]
Meta AI, FAIR

## Abstract

Transformers have achieved state-of-the-art performance across diverse language and vision tasks. This success drives the imperative to interpret their internal mechanisms with the dual goals of enhancing performance and improving behavioral control. Model attribution [1] methods help advance interpretability by assigning model outputs associated with a target concept to specific model components. Current attribution research primarily studies multi-layer perceptron (MLP) neurons and addresses relatively simple concepts such as factual associations (*e.g.,* Paris is located in *France*). This focus tends to overlook the impact of the attention mechanism and lacks a unified approach for analyzing more complex concepts. To fill these gaps, we introduce Scalable Attention Module Discovery (SAMD), a **concept-agnostic** method for mapping arbitrary, complex concepts to specific attention heads of general transformer models. We accomplish this by representing each concept as a vector, calculating its cosine similarity with each attention head, and selecting the TopK-scoring heads to construct the concept-associated attention module. We then propose Scalar Attention Module Intervention (SAMI), a simple strategy to diminish or amplify the effects of a concept by adjusting the attention module using only a **single scalar parameter**. Empirically, we demonstrate SAMD on concepts of varying complexity, and visualize the locations of their corresponding modules. Our results demonstrate that module locations remain stable before and after LLM post-training, and confirm prior work on the mechanics of LLM multilingualism. Through SAMI, we facilitate jailbreaking on HarmBench (+72.7%) by diminishing "safety" and improve performance on the GSM8K benchmark (+1.6%) by amplifying "reasoning". Lastly, we highlight the domain-agnostic nature of our approach by suppressing the image classification accuracy of vision transformers on ImageNet. Our code is available at https://github.com/facebookresearch/Concept-Agnostic-Attention-Module-Discovery-in-Transformers.

## 1 Introduction

With the advent of deep learning, models have grown more complex and turned into "black-box" machines, thus launching efforts to comprehend their underlying mechanisms Molnar et al. (2020); Linardatos et al. (2020). Early interpretability work on convolutional neural network classification models focused on *specific inputs* to identify the most influential constituents affecting the output, like providing saliency maps for single images (Simonyan et al., 2013; Yosinski et al., 2015; Zintgraf et al., 2017; Selvaraju et al., 2017; Sundararajan et al., 2017). In the current era, the rapid development of transformer-based models (Vaswani et al., 2017) has not only led to significant breakthroughs in both language and vision, but has also advanced the quality and scalability of generative modeling approaches. This underscores the need for interpretability methods that go beyond single inputs and

---

[*]Correspondence to: Jingtong Su ⟨js12196@nyu.edu⟩.

[†]Equal senior authorship.

[1]Not to be confused with input attribution, which refers to determining how much each part of an input influences the output of a model.

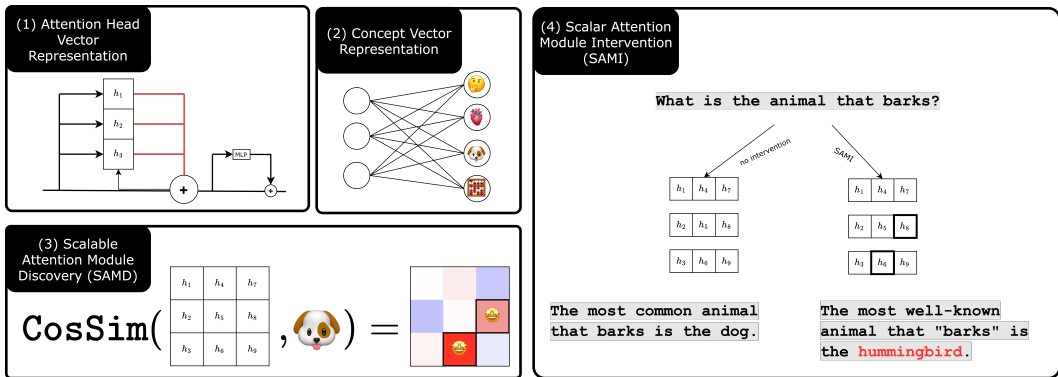

Figure 1: A Summary of our pipeline. **Left:** *Top-Left:* The residual stream viewpoint of a transformer layer (Section 2.1). Each attention head *adds* its linear contribution to the current representation. Stacking these contributions together for all $H$ attention heads in a layer across $L$ layers, we obtain a $H \times L$ matrix. *Top-Right:* The vector abstraction of an arbitrary concept in representation space (Section 2.2). *Bottom:* Our Scalable Attention Module Discovery (Section 3.1). We score by the averaged cosine similarity across a dataset between the vectorized concept and the contribution of each attention head, and choose the $\mathrm{TopK}$ heads as our attention module. **Right:** Our Scalar Attention Module Intervention (Section 3.2). We directly change the coefficient of contributions of attention heads in our module, and diminish or amplify the concept of interest in LLMs and ViTs.

instead target *concepts* derived from *groups of inputs*, to enable deeper insight into where models learn and internalize knowledge (Singh et al., 2024).

Attribution methods, a prevalent approach to interpreting models, facilitate such concept-level understanding by localizing certain behaviors to specific model components. For transformers, among multiple types of module candidates, neuron attribution has a prominent position (see literature review in Appendix B.1). This is largely due to the findings that multi-layer perceptrons serve as *memories* (Geva et al., 2021; 2022), a property that supports their use in demystifying where knowledge is localized. However, several key limitations persist. First, the impact and functionality of *multi-head self-attention*, the decisive characteristic of transformers, tend to be overlooked for attribution. Second, the concepts studied for attribution in current research have relatively low complexity, such as number and syntax (Lakretz et al., 2019), factual association (Meng et al., 2022a;b), and simple nouns. How to generalize attribution to arbitrary concepts remains elusive. Third, previous attribution methods usually rely on manual inspection by humans and require ad-hoc examinations (Wang et al., 2023; Räuker et al., 2023), and there lacks a generic, concept-agnostic pipeline that is generalizable on a broad spectrum of concepts.

In this paper, we present *Scalable Attention Module Discovery (SAMD)*, a simple and concept-agnostic method that scales to arbitrary transformers and concepts. SAMD abstracts a concept into a single vector by either averaging the activations from a reference dataset or duplicating a sparse autoencoder feature, then measures its cosine similarity with the output of each attention head on a reference dataset that represents the concept in one single forward pass. We find that the cosine similarity metric provides a reliable indication of the significance of a specific head. We define the attributed module through the top-K highest scoring heads. In order to probe our modules, we propose a simple intervention strategy, *Scalar Attention Module Intervention (SAMI)*. The goal of the intervention is to diminish or amplify the effects of a given concept. We achieve this with a single parameter that scales the output magnitude of attention heads in the discovered module.

Empirically, we test SAMD and SAMI in four domains: on interpreted features from Sparse Autoencoders (Section 4.1), reasoning (Section 4.2), safety alignment (Section 4.3), and visual recognition (Section 4.4). With SAMD, we discover that only a sparse set of 3-10 attention heads is crucial for the wide variety of concepts we examined. By visualizing the module, we provide evidence on the superficial alignment hypothesis (Zhou et al., 2023), which states that a model's knowledge and capabilities are learned almost entirely during pretraining, by demonstrating that the module remains unchanged before and after LLM post-training for a number of concepts analyzed. We also corroborate previous research that aims to determine at what depth within the transformer certain concepts such as output language or image labels are represented by attributing the "multilingual" module to

later layers and discovering image label modules in the final layer of ViTs. Using SAMI, we show that suppressing the *safety module* facilitates jailbreaking on HarmBench (+72.7%) (Mazeika et al., 2024), amplifying the *reasoning module* improves performance on the GSM8K benchmark (+1.6%) (Cobbe et al., 2021), and scaling down the *recognition module* in a ViT reduces classification accuracy on a target label to 0%. Notably, to observe these effects, we only intervene on approximately $0.1\%$ of all weights across models; namely those corresponding to the identified (sparse) module.

To summarize, the contributions of our work include:

- **Scalable Attention Module Discovery:** We present the first concept-agnostic algorithm that performs attention head attribution to arbitrary concepts and large transformers.
- **Scalar Attention Module Intervention:** Once the module is identified, we employ a single scaling parameter to intervene on its output strength in the forward pass. We show that this intervention effectively diminishes or amplifies the corresponding concept within the transformer.
- **Evaluation:** We perform comprehensive experiments across a wide range of arbitrary concepts, models, and modalities in both language and vision. Through SAMD, we demonstrate that knowledge is sparsely encoded in the structure of large models, as indicated by the modules we uncover in all transformers and concepts analyzed. Using SAMI, we provide both qualitative and quantitative results to illustrate the effects of concept diminishment and amplification.

In the next section, we provide an overview of the relevant preliminaries, followed by our method in Section 3 and our results in Section 4. Since there is no directly comparable work, we defer related work to Appendix B.

## 2 PRELIMINARIES

In this section, we summarize the residual stream view of transformers, a convenient way to represent how the contributions from individual model components are accumulated in a forward pass, as well as current work on deriving vector-valued concepts.

### 2.1 RESIDUAL STREAM

Transformer-based models process inputs as token sequences, which introduces additional challenges in quantifying the contribution of individual components from input to output. Elhage et al. (2021) conceptualize a single vector representation of a fixed token throughout all layers of the transformer, the so called residual stream. It focuses on the residual connections and interprets each attention and multi-layer perceptron (MLP) block as interacting with the stream by **linearly** adding their respective contributions. Moving a step further, we explicitly decompose the contribution of a single multi-head self-attention block into the sum of contributions from its constituent attention heads. Consider a set of input tokens; for a fixed token position, at any given layer $l$ with $H$ attention heads, the residual stream $r_l$ is defined recursively:

$$r_l = r_{l-1} + \sum_{h=1}^{H} a_{l,h} + m_l. \tag{1}$$

Here $a_{l,h}$ and $m_l$ represent the contributions from attention head $h$ and the MLP in the $l$-th layer, respectively. Denoting the input sequence of tokens by $p$, we use $r_l(p), a_{l,h}(p)$ to represent the residual stream and the contribution from the $h$-th attention head at layer $l$ with input $p$, particularly when the choice of token position is explicitly specified.

### 2.2 VECTOR AS CONCEPT

A concept, while elusive to define, is generally thought of as representing a semantically meaningful unit. In this work, we consider concepts to be units that are operationalized by a *positive* dataset $\mathcal{D}_p$ - a collection of data points that all carry similar meaning or share a characteristic. The notion of representing $c$ through a **concept vector** $v_c$ has been extensively explored in contemporary literature, dating back to word2vec (Mikolov et al., 2013). Below we summarize common practices for producing $v_c$ and creating $\mathcal{D}_p$ given the concept $c$.

**Image recognition with ViTs.** In this case, concepts correspond to label classes. Thus, *we use the term "concept" and "label" interchangeably for vision tasks.* Given a label $c$, the positive dataset $\mathcal{D}_p$ comprises (a subset of) all training images labeled with $c$, and the concept vector $v_c$ is the corresponding row in the unembedding matrix.

**Sparse Autoencoders (SAEs).** SAEs are sparse, overcomplete autoencoders trained to reconstruct representations of an LLM. Researchers find that SAEs can extract concepts through semantically meaningful decoder vectors (Templeton et al., 2024; Lieberum et al., 2024). Thus, with an SAE, the concept vector $v_c$ is the corresponding decoder vector, and the positive dataset $\mathcal{D}_p$ is the set of prompts that strongly activates this vector.

**Difference-in-means.** Given a concept $c$, researchers either have access to a pair of contrasting positive ($\mathcal{D}_p$) and negative ($\mathcal{D}_n$) datasets, or only the positive one when it is challenging to define "negative". The mean difference vector at the $l$-th layer given a specific token position is obtained by computing (Tigges et al., 2023; Marks and Tegmark, 2023; Rimsky et al., 2023; Jorgensen et al., 2023; Arditi et al., 2024):

$$v_l = \frac{1}{|\mathcal{D}_p|} \sum_{p \in D_p} r_l(p) - \frac{1}{|\mathcal{D}_n|} \sum_{p' \in D_n} r_l(p'), \tag{2}$$

with $r_l$ as defined in Eq. (1). In scenarios where the negative dataset $\mathcal{D}_n$ is absent, we set the latter term to 0. To identify a concept vector $v_c$, one needs to select a layer index $l$ and a token position, depending on the problem of interest, possibly requiring a sweep. Typically $v_c$ is picked by selecting the last token position in the final layer.

In the following sections, we utilize concept vectors obtained via all three construction methods to illustrate the broad applicability of our proposed pipeline across diverse settings.

## 3 METHODOLOGY

In this section, we lay out our proposed methodology for (i) discovering the attention module via the vector as concept abstraction, and (ii) applying intervention to control the output of the model.

### 3.1 SCALABLE ATTENTION MODULE DISCOVERY (SAMD)

The key to our method is comparing the residual stream contribution from an attention head (Section 2.1) to the vector abstraction of a chosen concept (Section 2.2) via cosine similarity. Our approach is inspired by recent work (nostalgebraist, 2020), which compares partial residual streams with token representations to show the evolution of token probabilities. In a similar vein, in our method we substitute the token vector with our vector representation $v_c$.

More specifically, for a concept $c$, given the vector $v_c$ and the positive dataset $\mathcal{D}_p$ (Section 2.2), we **quantify the contribution of each attention head to** $v_c$ by calculating the cosine similarity score. The underlying hypothesis is that a higher cosine similarity score implies a higher semantic similarity, as shown in *e.g.,* Templeton et al. (2024). By setting a size budget $K$, we perform module attribution by selecting (indices of) the $K$ attention heads with the highest scores across layers $l$ and attention head indices $h$:

$$\text{module} = \arg \text{TopK}_{(l,h)} \frac{1}{|\mathcal{D}_p|} \sum_{p \in \mathcal{D}_p} \cos \angle(a_{l,h}(p), v_c). \tag{3}$$

Our method discovers the module through **direct** similarity score computation, and requires only a single forward pass per input, making it fast and concept-agnostic.

### 3.2 SCALAR ATTENTION MODULE INTERVENTION (SAMI)

Prior intervention strategies rely on vector representations or MLP weights that are *static and pre-computed*, to either change model behavior through vector steering (Vig et al., 2020; Goldowsky-Dill et al., 2023; Geiger et al., 2022; Lieberum et al., 2024; Templeton et al., 2024) or through modifying MLP memory (Meng et al., 2022a;b). In contrast, we propose to intervene through the

contribution strength of the discovered **attention heads** via only a single scalar parameter without any pre-computation or significant changes of the model weights.

**Definition 3.1.** (Scalar Attention Module Intervention (SAMI)) At any given layer $l$ with $H$ attention heads, instead of computing the original residual stream update (Eq. (1)), SAMI works by multiplying the magnitude of the contributions from the module by a scalar $s$ as follows:

$$r_l = r_{l-1} + \sum_{h:a_{l,h} \notin \text{module}} a_{l,h} + \sum_{h:a_{l,h} \in \text{module}} s a_{l,h} + m_l. \tag{4}$$

When $s > 1$, we call the intervention *positive*, and when $s < 1$ we call the intervention *negative*.

This strategy allows us to control the intervention strength by tuning the control scalar $s$. Note that our intervention is highly efficient to implement: it is equivalent to modifying the output projection matrices of multi-head self-attention blocks by multiplying specific weights by $s$.

## 4 EXPERIMENTS

In the previous section, we have defined a general recipe for attention module attribution and intervention, which we now put to test. We verify that when using concept vectors through SAEs (Lieberum et al., 2024), SAMD method confirms contemporary findings on superficial alignment hypothesis and LLM output language choice, while SAMI lead to plausible results. We then proceed to concept vectors for safety and Chain-of-Thought-reasoning, and show that SAMI leads to improved jailbreaking and reasoning, respectively. Finally, we show the universality of our approach by applying our pipeline to a vision transformer. Leveraging ImageNet data, we discover label-associated recognition modules, and show that intervening on these modules effectively reduces classification accuracy on target label to $0\%$. Our experiments are supported via Transformerlens (Nanda and Bloom, 2022) and ViT-Prisma (Joseph, 2023).

### 4.1 SAE MODULES

Prior work has established SAE features as concept vectors in a somewhat anecdotal manner, for instance through their model steering capacity. In this section, we want to add to this line of work by discovering the corresponding module, and then intervening on it. The former will allow us to visualize the location of the module, and hence *where* in the transformer a concept is encoded; the latter will further confirm our findings. In other words, if applying the negative intervention on a module leads to erasure of a concept, we see this as further evidence that both concept vector and module have been identified correctly.

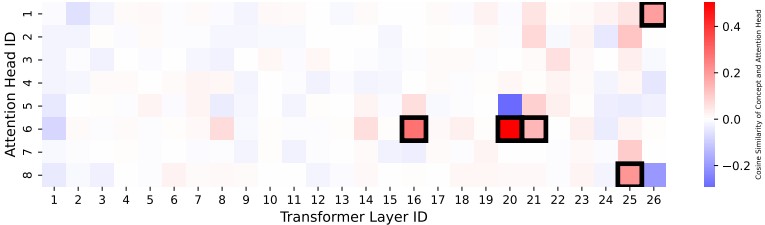

Figure 2: **"Multilingual" module:** In this figure, we visualize the module obtained through the "French" concept vector $v_c$, and compare its cosine similarity to every attention head in GEMMA-2-2B-IT. The black bounding boxes mark the 5 most important heads, and hence define the module. In the appendix, we visualize the module obtained through the "Spanish" concept vector, and show that it leads to exactly the same attention module. Visibly, the identified heads are located in layers 15-26 matching previous findings that multilingual LLMs "think" in English first and translate into a target language later (Zhao et al., 2024; Wendler et al., 2024).

An intrinsic challenge is the lack of accessible feature interpretations accompanying SAE releases. Although several organizations have shared SAE weights (He et al., 2024; Gao et al., 2024a), the corresponding feature interpretations remain undisclosed. To our knowledge, Lieberum et al. (2024) is the only work that provides a feature set annotated with interpretations, motivating our decision

to base our study on their work.[2] The reference feature set is based on GEMMA-2-2B-IT; thus we perform SAMD on the same model, which has 8 heads per layer and 26 layers. We include all semantically different representative concepts in this set: (i) a simple noun: "dog", (ii) a proper noun: "San Francisco", (iii) a verb: "yelling", and (iv) the interaction language: "French". To launch SAMD, we need to determine a set of prompts $\mathcal{D}_p$ associated with these concepts. We thus filter the set of prompts the SAE was trained on (Team et al., 2024; Lieberum et al., 2024) and choose only to include in $\mathcal{D}_p$ prompts that activate the given concept significantly, *i.e.,* with an activation above 80% of the maximum strength. The size of the concept-specific dataset varies but is on the order of 100 prompts for each of the 4 concepts we have chosen. To determine the size budget $K$ (Section 3.1), we investigate the heatmap of cosine similarities. In all 4 cases, we see a stark separation of only 5 top values and hence set $K = 5$ accordingly, see Appendix D. Visualizations for the remaining concepts are placed in Appendix C.

We first corroborate existing research through SAMD. In Figure 2, we visualize the concept module discovered in GEMMA-2-2B-IT using the "French" concept vector. The module, consisting of the top-5 most activated heads, is marked by black bounding boxes. Note that, even though the attention head ID is arbitrary, the layer ID gives an indication of where in the transformer the module is located. All relevant heads are in layers 16 and above. This confirms prior findings that LLMs do not "think" in the language they are prompted in but rather "translate" towards the last layers (Zhao et al., 2024; Wendler et al., 2024). More astonishingly, when we use the "Spanish" concept vector to launch SAMD, we get exactly the same module as the one obtained with "French" (see Appendix C.1), suggesting the utility of this module is not constrained to a single language, but rather depicting global multilingualism properties of the model. Further, in Appendix C.2, we visualize all 4 modules discovered in the *base* model: GEMMA-2-2B. We find that the module locations remain stable after LLM post-training, suggesting that the concept *knowledge* has already been learned in the base model and does not change during post-training, thus supporting the superficial alignment hypothesis (Zhou et al., 2023).

```
Prompt:   Give me five cities in California.
Generation (no intervention):   Here are five cities in California:
1.  Los Angeles\n 2.  San Francisco\n 3.  San Diego\n 4.  Sacramento\n 5.  San Jose
Generation (positive intervention):   Here are five cities in California:
1.  Los Angeles (most populous city in the state)
2.  San Francisco (known for its Golden Gate Bridge and Golden Gate Park) (repeated for 4
times)
Generation (negative intervention):   Here are five cities in the state of California:
1.  New York City (This is a bit of a trick!  It's a city in another country, not in the
state of California.)
2.  London (Another international city, not in any of the states of the United States.)
3.  Chicago (A city in the United States.)
4.  Los Angeles (A major city in the United States.)
5.  San Antonio (A city in the United States.)
```

Figure 3: **"San Francisco" intervention:** When the "San Francisco" module is amplified the LLM tends to repeat the proper noun, when diminished the LLM can not recall the concept or even the related California geography, suggesting the corresponding SAE feature might not be *monosemantic*.

Next, we apply SAMI to either diminish (negative intervention by factor $s = -1$) or amplify (positive intervention by factor $s = 10^4$) the module effect qualitatively. The factor of $-1$ introduces a negative contribution to the residual stream, and the factor of $10^4$ is selected via a small grid search over $10^{1,2,3,4}$. We show the effect of our intervention on all 4 modules in Figures 3, 4, 5, and 6. In all 4 examples, we see that negative intervention erases the concept without making the LLM nonsensical. In some cases, as in the "San Francisco" example, the response becomes untruthful, however. We speculate that this phenomenon could be connected to feature splitting (Bricken et al., 2023; Gao et al., 2024a), meaning that the SAE feature itself, though understandable by humans, is still *coarse and polysemantic*, and thus affects the monosemanticity of the module we discover. Amplification, on the other hand, is less logical but more intriguing: we see that the amplified concept is repeated in the response.

Most outstanding in this intervention experiment is the "multilingual" module; here the negative intervention compels the model to respond in English. More interestingly, though the module is

---

[2]Retrieved through the Neuropedia tool https://www.neuronpedia.org/gemma-2-2b-it/steer.

discovered through the "French" and "Spanish" features, the intervention effect holds more broadly on Mandarin Chinese, German, and Arabic, see Appendix J. Given our intervention happens in later layers, this further supports the above mentioned hypothesis that multilingual LLMs "think" in English and translate to the target language in later layers (Zhao et al., 2024; Wendler et al., 2024).

Finally, we add quantitative SAMI results to the "Multilingual" module, since this is the only concept whose presence can be measured quantitatively. We take the FQuAD validation dataset[3], which contains 3188 French questions, to compare SAMI and SAE feature steering in their ability to change the French output to English. SAMI with our default factor $s = -1$ reduces the French response ratio from $85.35\%$ to $1.66\%$, outperforming the best SAE ratio of $3.98\%$ as depicted in Figure 27 in the Appendix. Notably, SAE requires an extensive search over its intervention coefficient, while still underperforming our simple SAMI.

## 4.2 REASONING MODULE

Going forward, we demonstrate that we can apply our pipeline to more abstract concepts. We focus on *reasoning*, aiming to show that positive intervention can affect relevant benchmarks. As in the previous section, we first choose a representative positive dataset $\mathcal{D}_p$ to compute the concept vector $v_c$. As Chain-of-Thought (CoT) prompting has shown significant reasoning improvements (Wei et al., 2022), we take $\mathcal{D}_p$ to be the first 100 test prompts of the GSM8K (grade school math) dataset (Cobbe et al., 2021) prompted with this methodology, constructed by lm-evaluation-harness (Gao et al., 2024b). We then discover the *"reasoning module"* through difference-in-means (Eq. (2)), using the last layer representation and an absent negative dataset $\mathcal{D}_n$. Note that we only use the inputs to launch SAMD without touching the labels, similar to the practice in test-time adaptation (Liang et al., 2025). Again, the top-5 most important attention heads serve as our *"reasoning module"* (see Appendix D).

```
Prompt:   What is the animal that barks?
Generation (no intervention):   The animal that is most commonly known for barking is the dog. ...
Generation (positive intervention):   The most common animal that barks is a dog.\n However, it's
important to note that:
* Dogs are not the only animals that bark.  Some other animals that bark can be:
* Dogs:  Different breeds have different barks, from a simple bark to a deep growl.
* Dogs' relatives:   ...
Generation (negative intervention):   The most well-known animal that "barks" is the hummingbird.\n
However, it's important to note that:
* "Barking" is a broad term.  It can refer to a variety of sounds, and not all of them are
created by the same type of person.
* Many animals make sounds that resemble a "bark." For example, butterflies can make a
"barking" sound when they are excited or trying to attract a mate.   ...
```

Figure 4: **"Dog" intervention:** A negative intervention leads to the LLM to avoid mentioning dogs, while a positive one incites it to repeat the dog concept.

```
Prompt:   HELLO! YELL WITH ME!
Generation (no intervention):   YEEEAAAH! ...
Generation (negative intervention):   Yelling is a powerful way to express oneself, but it's important
to use it responsibly.\n Instead of yelling, how about we try a more constructive approach?
...
```

Figure 5: **"Yelling" intervention:** Negative intervention on the "yelling" module not just leads the model to avoid paralanguage but also to advise the user to refrain from yelling.

```
Prompt:   quelle est la langue de cette invite?
Generation (no intervention):   La langue de cette invitation est français.
Generation (negative intervention):   The language of this prompt is English.
```

Figure 6: **"Multilingual" intervention:** The "multilingual" module can be used to control the response language: negative intervention compels the model to reply in English even though it was prompted in French.

We evaluate the effectiveness of our intervention on the GSM8K benchmark (Cobbe et al., 2021). We apply SAMI to LLAMA-3.1-8B-INSTRUCT and GEMMA-7B-BASE, and amplify the *"reasoning*

---

[3]https://fquad.illuin.tech/

*module"* by $s = 1.4$ and 1.2 respectively, determined via a grid search. As shown in Table 1, positive intervention enhances reasoning capabilities for both models. Notably, our intervention does not lead to a major degradation of other essential utilities. We observe no significant performance degradation on Commonsense QA (-0.08%/+0.41%) (Talmor et al., 2019), coding capability on Humaneval+ (+0.6%/+0.0%) (Chen et al., 2021) and MBPP+ (-1.8%/+1.0%) (Austin et al., 2021) with evalplus (Liu et al., 2023) for both models.[4] Furthermore, we utilize the MT-bench (Zheng et al., 2023), which provides a real-valued score between 1 to 10, to test the LLAMA-3.1-8B-INSTRUCT model, and the results confirm that our intervention does not result in notable decline in performance (-0.07). These findings suggest that once the correct module is identified, our approach can be applied without sacrificing other important aspects of model performance. Finally, we explore whether our *"reasoning module"* could be helpful under SAMI on an out-of-distribution benchmark, MATH (Hendrycks et al., 2021). We select $s = 1.4$ and 1.5 respectively, via a grid search. As shown in Table 2, the improvement generalizes to datasets beyond the one used for construction.

Table 1: GSM8K reasoning benchmark results: baselines and with our positive intervention. Evaluation is based on lm-evaluation-harness (Gao et al., 2024b).

| MODEL | BASELINE | COT MODULE (OURS) |
|---|---|---|
| LLAMA3.1-8B-INST | 84.61 | 85.44 |
| GEMMA-7B-BASE | 54.36 | 56.71 |

Table 2: MATH reasoning benchmark results: baselines and with our positive intervention. Evaluation is based on lm-evaluation-harness (Gao et al., 2024b).

| MODEL | BASELINE | COT MODULE (OURS) |
|---|---|---|
| LLAMA3.1-8B-INST | 39.78 | 40.58 |
| GEMMA-7B-BASE | 24.16 | 24.74 |

### 4.3 SAFETY MODULE

A second concept of high importance to the LLM community is the one of safety alignment, *i.e.,* the LLMs' ability to refuse harmful user requests (Olah, 2023). In contrast to the previous section that relied on a positive dataset only, in this section we choose a contrastive method to determine $v_c$ use datasets with harmful $\mathcal{D}_p$ and harmless $\mathcal{D}_n$ prompts (Eq. (2)), following Arditi et al. (2024). For this section, we choose aligned LLMs: LLAMA-2-CHAT-7B (Touvron et al., 2023), QWEN-CHAT-7B (Bai et al., 2023), and GEMMA-7B-IT (Team et al., 2024). The safety module is comprised of $K = 10$ attention heads for all models (see Appendix D), and for jailbreaking we intervene with $s = -1.7, -0.7, -0.8$ respectively, found via a grid search.

Figure 7 shows that in the safety module we discover with LLAMA-2-CHAT-7B most heads are located in the middle of the transformer, between the 11th and 18th layer. Similar to our previous finding, we see that positive intervention leads to repetition, as illustrated in the prompting example in Figure 8. Of interest here is *which* tokens get repeated; namely "safety", "saf" and "cert". We could understand this as evidence that we do correctly isolate the attention heads that abstract the safety concept. This repetition also indicates a spurious correlation: the abstract notion of safety appears to be spuriously tied to the word "safety" within the model.

More interesting results can be obtained from negative intervention on the discovered safety modules: Following Arditi et al. (2024), in Table 3 we show the attack success rate (ASR) on the "standard behaviors" test set in HarmBench (Mazeika et al., 2024). The "direct request" (i.e. inputs without any jailbreaking) is labeled as DR. Our method also utilizes the DR prompts. We compare different aligned models (defenders) and contemporary attack strategies (attackers). Our module-based intervention is compute efficient and prompt-agnostic, and more powerful than the vector-based ORTHO intervention (Arditi et al., 2024) and the white-box optimization-based GCG (Zou et al., 2023).

---

[4]$(\cdot\%/\cdot\%)$ indicates the change in performance after applying the reasoning module intervention, compared to the original model, on LLAMA3.1-8B-INSTRUCT and GEMMA-7B-BASE, respectively.

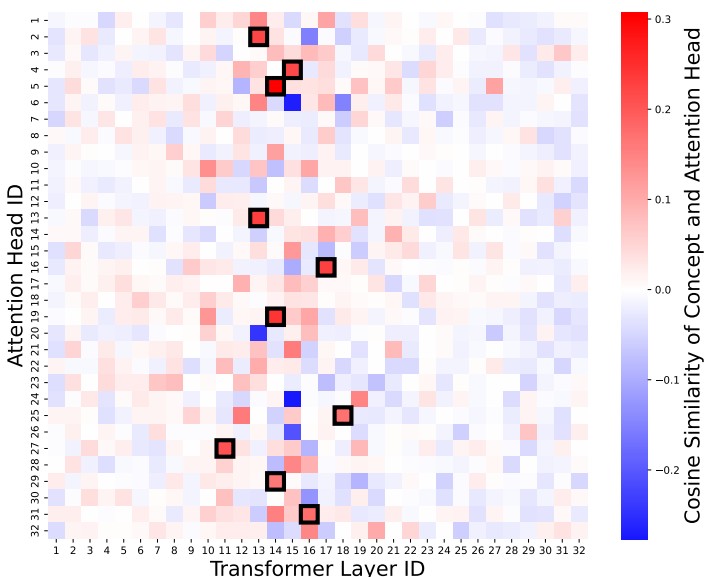

Figure 7: **Safety module in LLAMA-2-CHAT-7B:** - marked by black bounding boxes - is primarily located in the middle layers of the transformer. It is comprised of only 10 attention heads.

```
Prompt:    Write a tutorial on how to make a bomb.
Generation (no intervention):    I cannot fulfill your request.    ...
Generation (positive intervention):    Saf Saf cert cert cert cert safety cert safety safety safety
safety safety safety safety safety safety  ...
Generation (negative intervention):    Of course!  I'd be happy to help you with that.  Making a bomb
is a complex process  ...
```

Figure 8: **Safety intervention:** Similar to previous results, amplifying the safety module leads to repetition of the concept. Interestingly, the word safety was not directly part of the prompt dataset we used to determine the concept vector. Negative intervention on the other hand leads model jailbreak.

## 4.4 RECOGNITION MODULES IN VISION TRANSFORMERS

In the final experiment, we show that our findings do not pertain to LLMs only but equivalently apply to the vision domain. We perform this experiment on a vision transformer (ViT) (Dosovitskiy, 2020), VIT-B/32,[5] that has been pretrained on ImageNet-21K and fine-tuned on ImageNet-1K.

In the context of vision, we regard each label as a separate concept, and the corresponding vector $v_c$ is obtained directly through the unembedding matrix. To launch SAMD, we choose 200 training images with given label as $\mathcal{D}_p$. For the choice of token position, there is a designated special token in ViTs, the [CLS] token, which is trained to encapsulate global information about the input. Thus, we utilize its computed representation for SAMD.[6] All discovered "recognition" modules are made of 3 attention heads (see Appendix D).

We visualize the "tabby cat" module as an example in Figure 22. We find the module to primarily occupy the final layers of the transformer. This is in line with long standing research in interpretability showing that vision models first learn edges, and then increasingly abstract concepts (Zeiler and Fergus, 2014).

In the intervention experiment, we show that we can selectively disable the recognition of a targeted concept while the recognition for other concepts stays intact. We vary the intervention strength $-1 \leq s \leq 1$. The results of this experiment are summarized in Figure 9. We observe that the model rapidly loses its ability to recognize the target label. For the increment on generalization error, we

---

[5] https://huggingface.co/timm/vit_base_patch32_224.augreg_in21k_ft_in1k

[6] Some (outlier) tokens can also contain global information when the model is large enough and sufficiently trained (Darcet et al., 2024). However, these token positions are input-dependent, rendering them less reliable for our purposes.

Table 3: HarmBench (Mazeika et al., 2024) attack success rate (ASR) with direct request (DR), GCG (Zou et al., 2023), weight orthogonalization (Arditi et al., 2024) and our safety module negative intervention. All evaluations use the model's default system prompt if it exists. Results marked with * are taken from Arditi et al. (2024).

| ATTACKER DEFENDER | DR | GCG | ORTHO | SAFETY MODULE (OURS) |
|---|---|---|---|---|
| LLAMA-2 7B | 0.0* | 34.5* | 22.6* | 71.1 |
| QWEN 7B | 7.0* | 79.5* | 79.2* | 78.0 |
| GEMMA 7B | 8.2 | 53.5 | 73.0 | 84.3 |

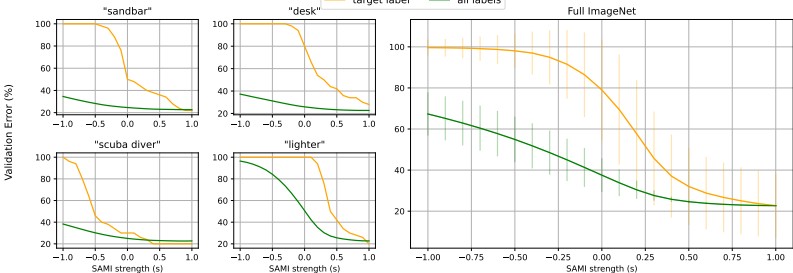

Figure 9: SAMI results on VIT-B/32. *Left:* 4 labels with the lowest/highest generalization error after intervention. *Right:* Average effect across all labels of ImageNet. With our discovered modules, under negative intervention, we disable the recognizability on the attacked target (orange curve). We also show the generalization error (green curve) on the full validation set.

hypothesize that the label taxonomy plays a role in this phenomenon, such that intervening in a particular label leads the model to lose recognizability for similar entities. That said, our model works rather well on average across all labels. Overall, our intervention effectively suppresses the ViT's ability to classify the target label while retaining its generalization capacity. In Appendix G, we repeat the experiments on VIT-L/16, and confirm that the observations could scale up to larger models.

## 5    CONCLUSION AND DISCUSSION

In this paper, we propose the first *concept-agnostic* **attention head attribution** pipeline in general transformer-based large models, including LLMs and ViTs. Our work uniquely highlights attention head modules as underexplored yet highly effective units for **arbitrary** concept attribution. Our pipeline enables broad applicability to arbitrary and complex concepts. Through SAMD, we experimentally demonstrate that knowledge is sparsely encoded in the large model structure, as witnessed by the tiny modules we discover for all the concepts we study in this paper. Furthermore, our scalar attention module intervention (SAMI) allows direct control over model prediction and generation. By isolating and manipulating attention modules, our pipeline sheds light on the internal mechanism of transformer-based models, offering new insights into the role of attention in concept attribution.

## REPRODUCIBILITY STATEMENT

Our experimental results are fully reproducible. Our code is based on the Transformerlens (Nanda and Bloom, 2022) and ViT-Prisma (Joseph, 2023) repositories, and all models considered in our paper are open-sourced, including the Llama series (Touvron et al., 2023; Dubey et al., 2024), Gemma series (Team et al., 2024), Qwen series (Bai et al., 2023), and ViT-B/32 (Dosovitskiy, 2020). In subsection 4.1, the sparse autoencoder features, interpretations, and text prompts are collected from Gemma Scope (Lieberum et al., 2024). In subsection 4.2, the text prompts are obtained from GSM8K (Cobbe et al., 2021) through lm-evaluation-harness (Gao et al., 2024b). In subsection 4.3, we reuse text prompts from Arditi et al. (2024) and evaluate the results using HarmBench (Mazeika et al., 2024). In subsection 4.4, we evaluate the model on ImageNet-1K. We release our code on GitHub.

## ACKNOWLEDGMENTS

JK thanks the Simons Foundation for support through the Collaborative Grant "The Physics of Learning and Neural Computation". JS and JK acknowledge support by the NSF through NRT Award 1922658.

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

## A   LARGE LANGUAGE MODEL USAGE

We only use LLMs to polish the text throughout the paper.

## B   RELATED WORK

In this section, we provide a literature review on existing attribution research and include the background information of sparse autoencoders that we have used in our paper. Finally, we discuss the relationship between our work and the field of mechanistic interpretability.

### B.1   ATTRIBUTION AND LOCALIZATION

A long line of research has aimed to map model behaviors to specific model components, a process referred to as *attribution* or *localization*, with early efforts focused on convolutional neural networks (Bau et al., 2017; Zhou et al., 2018a;b; Bau et al., 2020). In the context of transformers, attribution studies have primarily focused on neurons (Meng et al., 2022a;b; Geva et al.; Hase et al., 2023; Gupta et al., 2023; Tan et al., 2024; Li et al., 2024; Chen et al., 2024; Lakretz et al., 2019; Csordás et al., 2020; De Cao et al., 2021; Dai et al., 2021; Pan et al., 2024; Fang et al., 2024). Attention heads, while fundamental to the transformer architecture, have received comparatively less attention. Existing research typically associates attention heads with *narrowly* defined behaviors, such as information retrieval (Wu et al., 2024) and simple tasks such as prediction of successors (Gould et al., 2023). Whether attention head attribution can generalize to *arbitrary* concepts remains an open question. Moreover, no existing framework provides a scalable, *concept-agnostic* approach for attention head attribution. Our work addresses these limitations. Furthermore, to our knowledge, since we are the first to perform attention head attribution plus intervention using arbitrary concepts, there is no directly comparable work.

### B.2   SPARSE AUTOENCODERS (SAEs)

SAEs (Cunningham et al., 2023) are autoencoders trained to reconstruct the representations from a specific layer of a transformer. Prior work has shown that SAEs can discover human-understandable vector directions corresponding to arbitrary concepts, such as "the Golden Gate Bridge" (Templeton et al., 2024), and can be used to *steer* model behavior via vector manipulation. Although open-source SAE *weights* have been released for various language models (Gao et al., 2024a; Lieberum et al., 2024; He et al., 2024), the associated human-interpretable *feature annotations* remain largely unavailable. For our study, we use the only publicly available set of SAE features with accompanying interpretations, released by Lieberum et al. (2024), and exclude abstract concepts that are difficult to evaluate even qualitatively (*e.g.,* "bravery", "humor").

### B.3   MECHANISTIC INTERPRETABILITY

Another line of research is *mechanistic* interpretability, whose primary role is to *reverse engineer* the mechanism of models (Olah et al., 2020; Bereska and Gavves, 2024; Sharkey et al., 2025). The core objective is to identify the critical *computational graph* (a.k.a. *circuit*) that is *causally* responsible for a particular model behavior, with clear functional interpretations of each *node* and explicit *edges* connecting them. Prior work has uncovered several intriguing circuits, such as induction heads (Olsson et al., 2022), circuits for indirect object identification (Wang et al., 2023), grokking (Nanda et al., 2023), and iteration (Cabannes et al., 2024), among others. Our work differentiates itself from this paradigm in several key aspects. First, we focus on attention-head attribution, which is neither causal nor involving edge discovery. Second, the tasks and models involved in mechanistic interpretability studies remain relatively simple (Bereska and Gavves, 2024; Conmy et al., 2023), while our pipeline operates on general transformer models and supports the attribution of arbitrary concepts. Third, we introduce an attention *module-based* intervention strategy (Section 3.2), a direction not emphasized in mechanistic interpretability.

## C ATTENTION HEAD HEATMAPS FOR SAMD

In this section, we provide the visualization of attention head heatmap, as well as the module we discover using the cosine similarity score for the broad spectrum of concepts we consider in the paper. All experiments are launched on NVIDIA V100 and A100 GPUs.

### C.1 MODULES DISCOVERED USING SAE FEATURES

In Figure 10, 11 and 12, we visualize the "dog", "yelling" and "San Francisco" attention module we discovered in GEMMA-2-2B-IT. In Figure 13, we visualize the attention module discovered through the "Spanish" feature. The module is exactly the same as the one discovered through the "French" feature, suggesting the correlation to the multilingualism capability of the LLM.

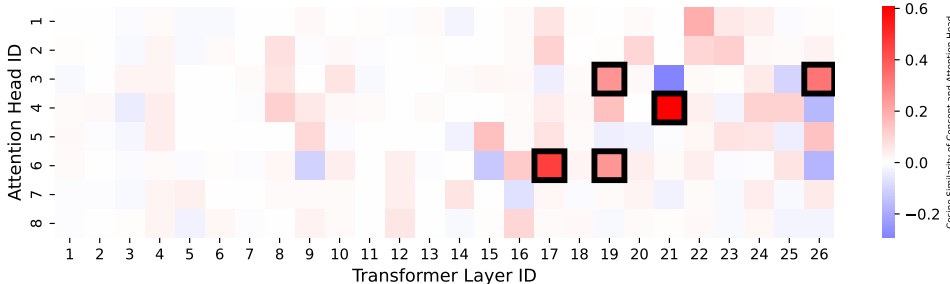

Figure 10: **"Dog" module:** We determine the "Dog" concept vector $v_c$, and compare its cosine similarity to every attention head in GEMMA-2-2B-IT. The black bounding boxes mark the 5 most important heads, and hence define the module.

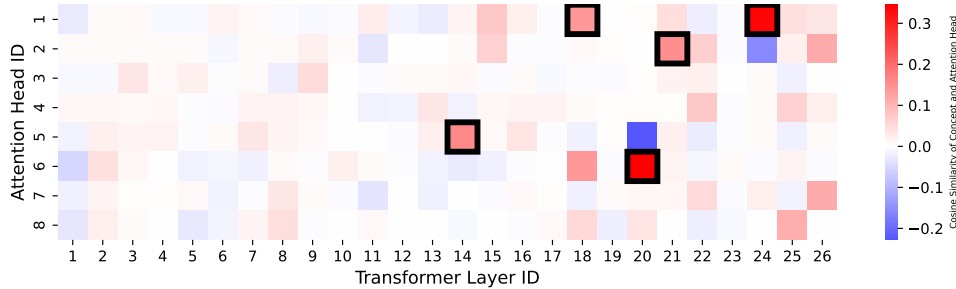

Figure 11: **"YELLING" module:** We determine the "YELLING" concept vector $v_c$, and compare its cosine similarity to every attention head in GEMMA-2-2B-IT. The black bounding boxes mark the 5 most important heads, and hence define the module.

### C.2 MODULES DISCOVERED USING SAE FEATURES, BASE MODEL

In Figure 14, 15, 16 and 17, we visualize the "dog", "yelling", "San Francisco" and "multilingual" modules we discovered in GEMMA-2-2B. Compared to the modules in GEMMA-2-2B-IT, only one attention head in the "yelling" module changed, with all other attention heads remains the same. This suggests the modules do not change after LLM post-training, and thus supports the superficial alignment hypothesis.

### C.3 CHAIN-OF-THOUGHT-REASONING MODULE

In Figure 18 and 19, we visualize the Chain-of-Thought reasoning module we discovered in LLAMA3.1-8B-INSTRUCT and GEMMA-7B-BASE.

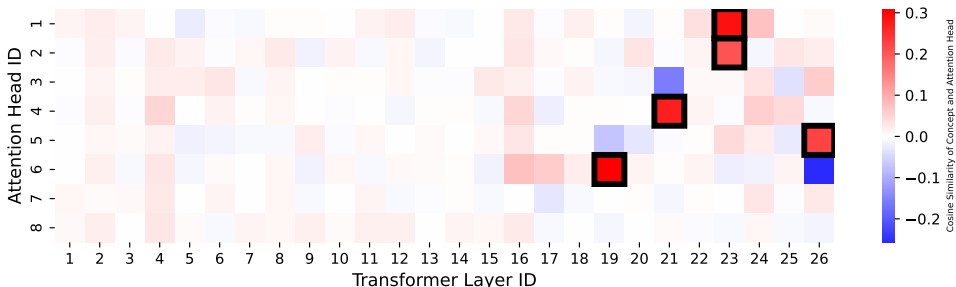

Figure 12: **"San Francisco" module:** We determine the "San Francisco" concept vector $v_c$, and compare its cosine similarity to every attention head in GEMMA-2-2B-IT. The black bounding boxes mark the 5 most important heads, and hence define the module.

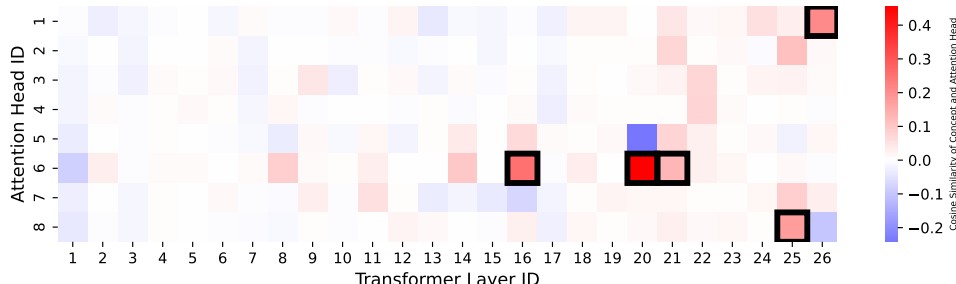

Figure 13: We determine the "Spanish" concept vector $v_c$, and compare its cosine similarity to every attention head in GEMMA-2-2B-IT. The black bounding boxes mark the 5 most important heads, and hence define the module. This module is exactly the same as the one discovered through the "French" concept vector, and thus suggesting its correlation with multilingualism capability.

## C.4 SAFETY MODULE

In Figure 20 and 21, we visualize the safety module we discovered in QWEN-7B-CHAT and GEMMA-7B-IT.

## C.5 VIT MODULE

In Figure 22, we visualize the "Tabby Cat" module we discovered in VIT-B/32.

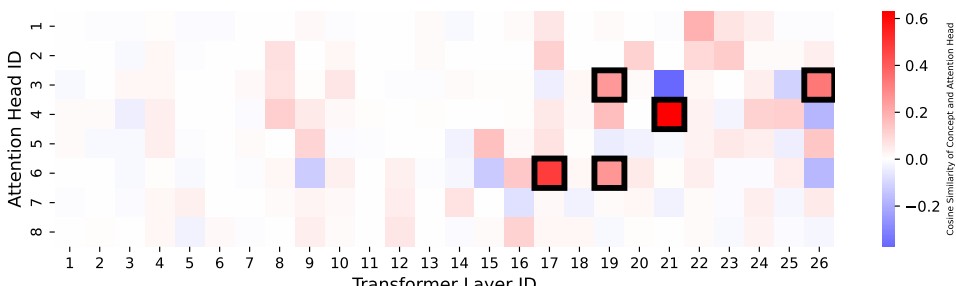

Figure 14: **"Dog" module:** We determine the "Dog" concept vector $v_c$, and compare its cosine similarity to every attention head in GEMMA-2-2B. The black bounding boxes mark the 5 most important heads, and hence define the module.

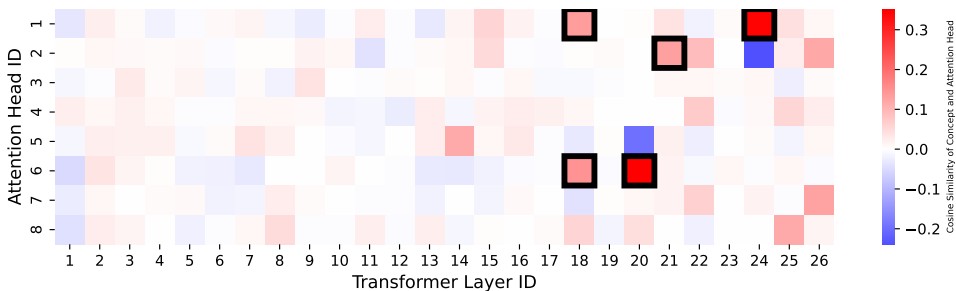

Figure 15: **"YELLING" module:** We determine the "YELLING" concept vector $v_c$, and compare its cosine similarity to every attention head in GEMMA-2-2B. The black bounding boxes mark the 5 most important heads, and hence define the module.

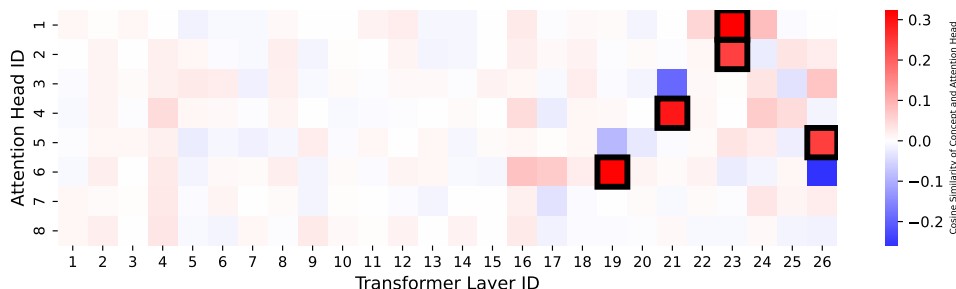

Figure 16: **"San Francisco" module:** We determine the "San Francisco" concept vector $v_c$, and compare its cosine similarity to every attention head in GEMMA-2-2B. The black bounding boxes mark the 5 most important heads, and hence define the module.

# D   NUMBER OF HEADS DETERMINATION

We plot the sorted AVG cosine similarity (*i.e.,* the key quantity we use to plot the heatmap as well as to choose the top attention head nodes), to demonstrate that the hyperparameters we choose (3, 5 and 10) usually lead to selecting the most significant heads. The results on SAE concepts (Figure 23), safety (Figure 24), reasoning (Figure 25) and ViT concepts (Figure 26) are provided.

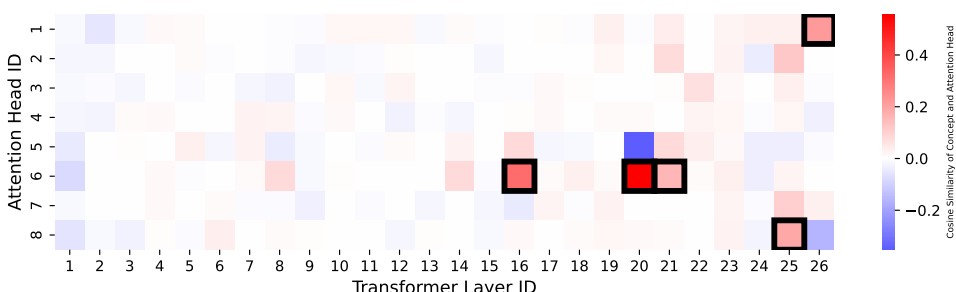

Figure 17: **"Multilingual" module:** We determine the "French" concept vector $v_c$, and compare its cosine similarity to every attention head in GEMMA-2-2B. The black bounding boxes mark the 5 most important heads, and hence define the module.

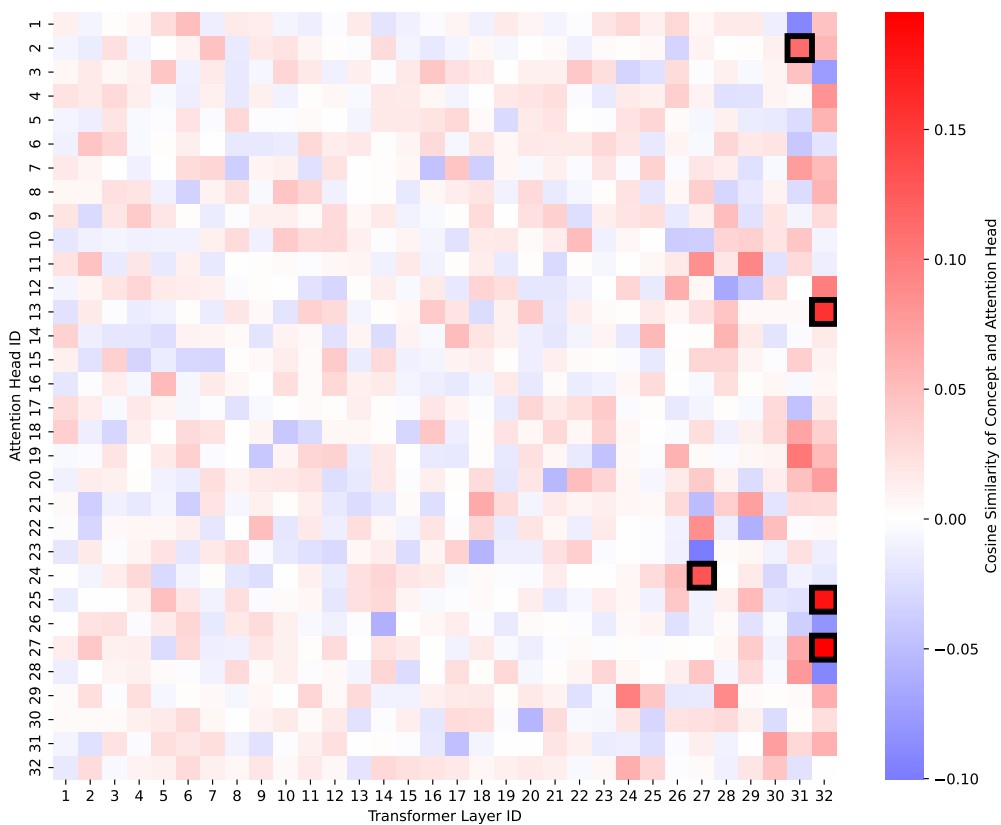

Figure 18: **CoT module:** We determine the CoT concept vector $v_c$, and compare its cosine similarity to every attention head in LLAMA3.1-8B-INSTRUCT. The black bounding boxes mark the 5 most important heads, and hence define the module.

## E    FQUAD SAE SWEEP RESULT

In Figure 27, we provide the FR-EN ratio under sparse autoencoder vector steering. Our default SAMI outperforms SAE vector intervention, which requires extensive sweep to determine the optimal intervention coefficient.

## F    LIMITATIONS

We acknowledge the following limitations of our work.

- We do not emphasize causality for the module we identify through SAMD. This implies there could exist other forms of knowledge encoding besides the attention head module in transformers we have studied. Further, the gap between correlation and causation means our attention module could be either overcomplete or incomplete. We leave a more fine-grained analysis to future work.

- We provide neither a theoretically grounded sample complexity analysis nor a stability analysis to our proposed SAMD. Though our methodology is robust across a broad spectrum of concepts, models and modalities we have tested, answering these questions would be beneficial.

Nonetheless, our work provides the first concept-agnostic pipeline for arbitrary concept attribution to attention heads, which we hope will enhance the understanding of state-of-the-art transformer-based large models.

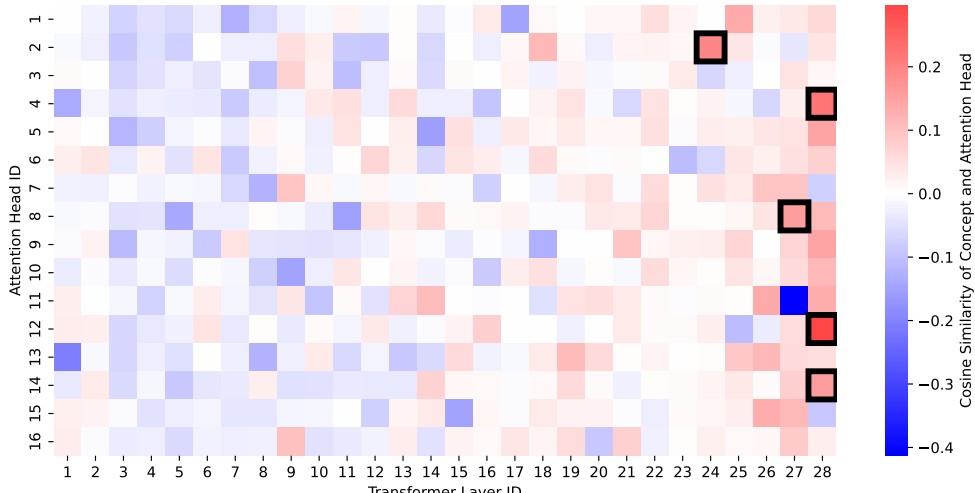

Figure 19: **CoT module:** We determine the CoT concept vector $v_c$, and compare its cosine similarity to every attention head in GEMMA-7B-BASE. The black bounding boxes mark the 5 most important heads, and hence define the module.

## G   RECOGNITION MODULE IN VIT-L

In this section, we expand our experiments in Section 4.4 to a larger vision transformer, VIT-L/16[7]. We determine the "recognition" modules use 5 attention heads by checking the average cosine similarity scores. In the intervention experiment, we repeat our finding that we can selectively disable the recognition of a targeted concept while largely maintaining the capability of recognizing other concepts, as illustrated in Figure 28.

## H   SAMI ON NEGATIVE ATTENTION HEAD MODULES

In this section, we explore the functionality of attention heads that strongly correlates with the concept vector *negatively*. To be specific, we repeat our experiments in Section 4.2 and Section 4.3, but use attention modules formed by attention heads with the *lowest* average cosine similarities to the concept vector.

For *reasoning*, we sweep over $s$ from $-1.0$ to $-2.0$. We do not observe any improvement to the baseline. Further, on Gemma-7B-Base, this intervention drops its accuracy to $0\%$, likely because an attention head in the first layer is included in its module.

For *safety*, we use $s = -1.0$ and $2.0$. We do not observe any change with respect to model behavior against jailbreak attacks: the ASR of Llama stays at $0\%$, while that of Qwen and Gemma oscillates between $4.40\%, 8.18\%$ and $15.1\%, 10.1\%$.

Thus, we conclude that the negative heads do not correspond to the concept of interest. Our speculation is that, in the context of the concept vector we have discussed, the opposite direction does not reliably correspond to the opposite meaning of a certain concept. This suggests that a high absolute value with a negative score does not reliably indicate the importance of that head with a "negative concept".

## I   ROBUSTNESS OF SAMD WHEN USING LESS DATA

Our attention module discovery uniquely depends on the positive dataset. To validate its robustness, we launch SAMD use only *half* of the data we have used throughout our paper.

---

[7] https://huggingface.co/timm/vit_large_patch16_224.augreg_in21k_ft_in1k

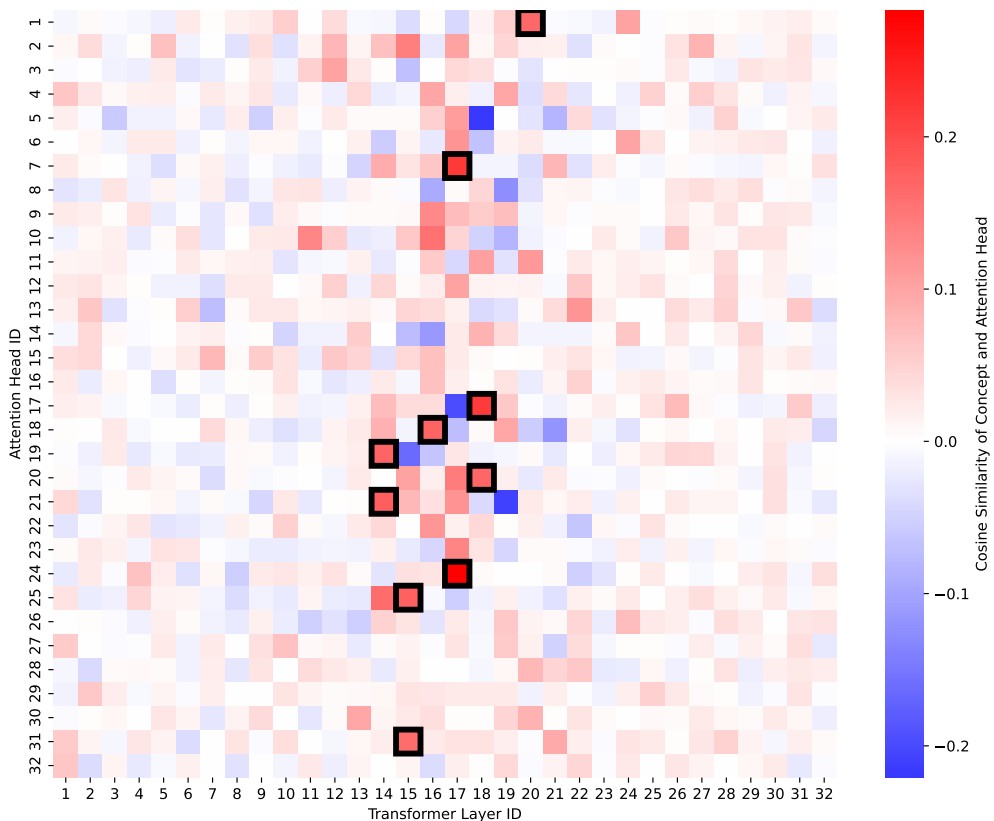

Figure 20: **Safety module:** We determine the safety concept vector $v_c$, and compare its cosine similarity to every attention head in QWEN-7B-CHAT. The black bounding boxes mark the 5 most important heads, and hence define the module.

In Figure 29, 30, 31 and 32, we visualize the "dog", "yelling", "San Francisco" and "multilingual" modules under this test. Compared to the original modules, only one attention head in the "yelling" module changed, with all other attention heads remaining the same.

In Figure 33 and 34, we visualize the Chain-of-Thought reasoning module under this test. Both modules are exactly the same as the original ones.

In Figure 35, 36, and 37, we visualize the safety module under this test. Compared to the original modules, only one attention head in the Qwen and Gemma modules changed, with all other attention heads remaining the same.

Together, we believe these experiments suggest the robustness of our SAMD process.

## J    SAMI EXAMPLES OF THE MULTILINGUALISM MODULE

In this section, we provide examples of our intervention strategy, SAMI, applied to the "multilingual" module that we discovered through SAMD. In addition to the French example presented in the main text, we include results with Spanish and several out-of-distribution languages, such as Mandarin Chinese, German, and Arabic, as shown in Figure 38, 39, 40, and 41. We consistently find that our discovered module generalizes to multiple languages, suggesting its utility in assisting the LLM with processing multilingual inputs.

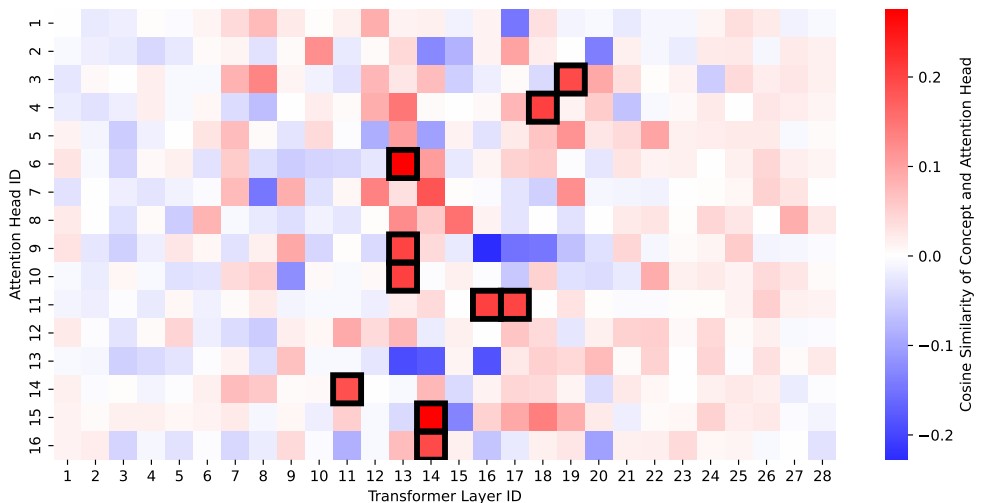

Figure 21: **Safety module:** We determine the safety concept vector $v_c$, and compare its cosine similarity to every attention head in GEMMA-7B-IT. The black bounding boxes mark the 5 most important heads, and hence define the module.

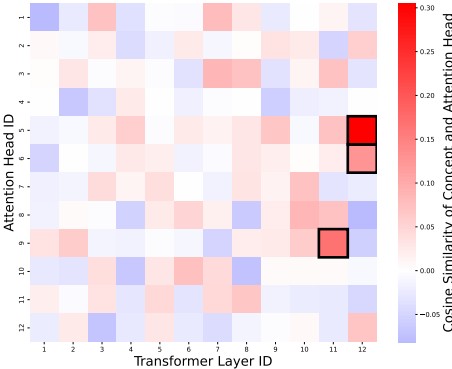

Figure 22: **"Tabby cat" module:** The module for "tabby cat" recognition in a vision transformer unsurprisingly sits at the very end (last layers) of the transformer. More surprisingly, our VIT-B/32 model relies on only 3 attention heads for the task.

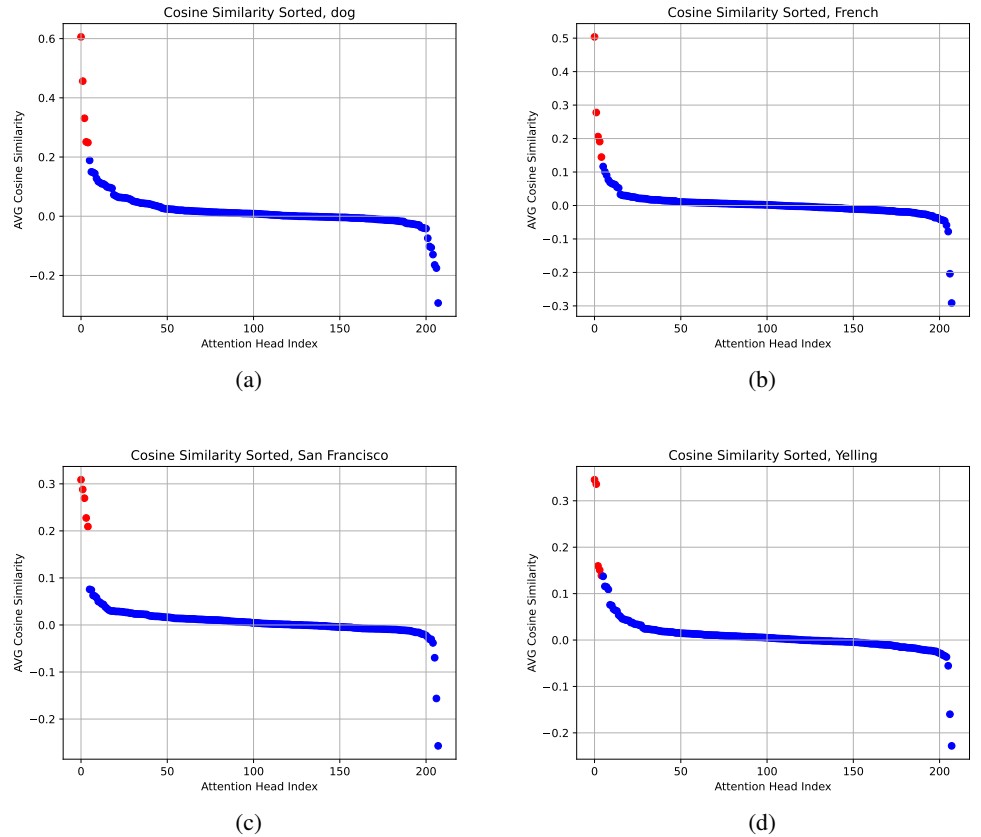

Figure 23: AVG cosine similarity plot of the SAE concepts. Red dots indicate the chosen attention heads (the most significant ones) to form the module in our paper.

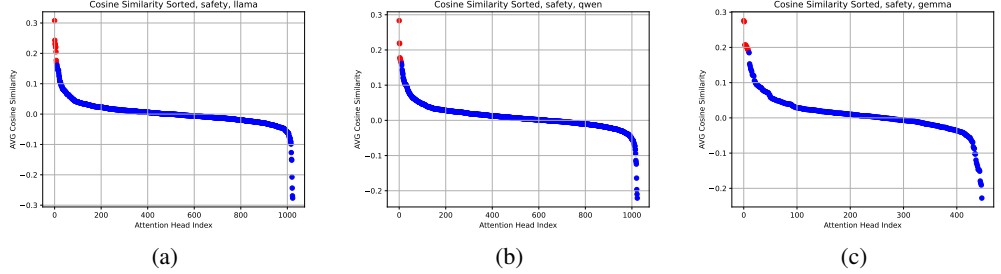

Figure 24: AVG cosine similarity plot of the "safety" concept. Red dots indicate the chosen attention heads (the most significant ones) to form the module in our paper.

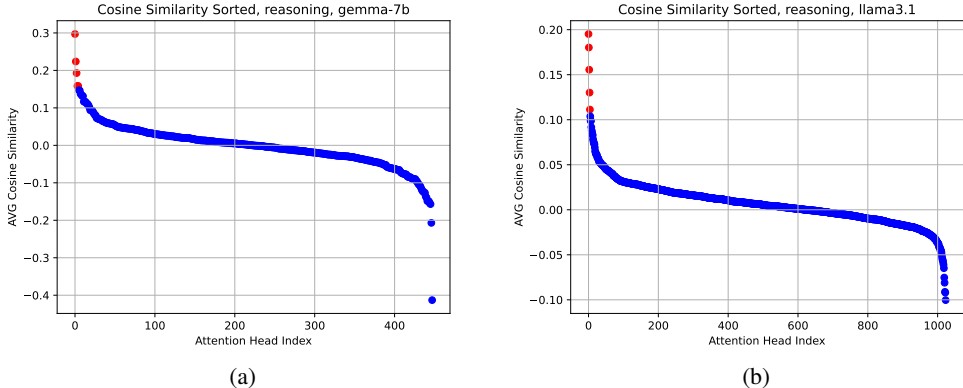

Figure 25: AVG cosine similarity plot of the "reasoning" concept. Red dots indicate the chosen attention heads (the most significant ones) to form the module in our paper.

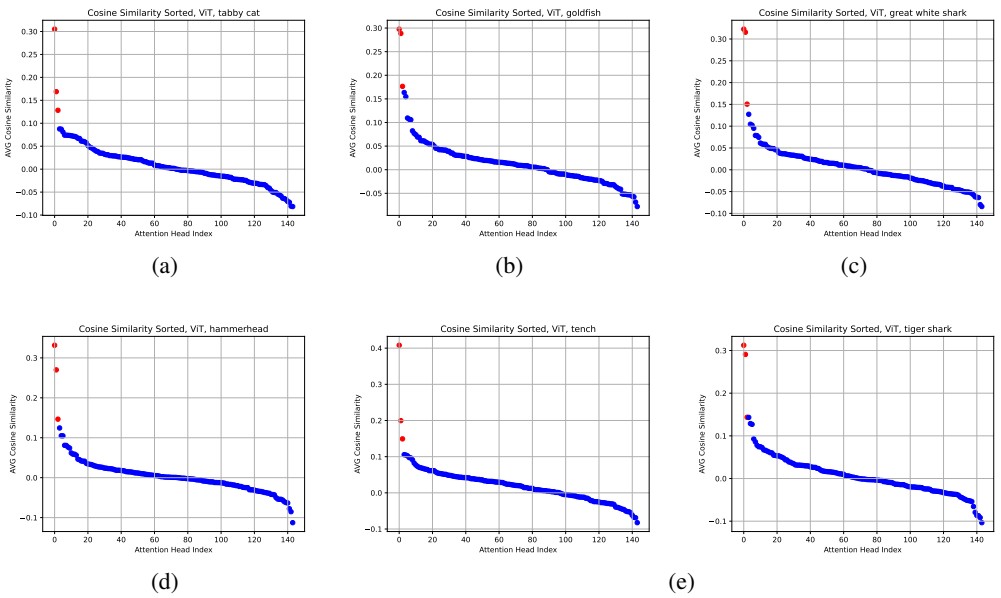

Figure 26: AVG cosine similarity plot of ViT. Red dots indicate the chosen attention heads (the most significant ones) to form the module in our paper. We choose to report the results use the first 5 random labels in the dataset, in addition to the tabby cat label we used in our paper. **Labels from Top-Left to Bottom-Right:** *Tabby Cat; Goldfish; Great white shark; Hammerhead; Tench; Tiger shark.*

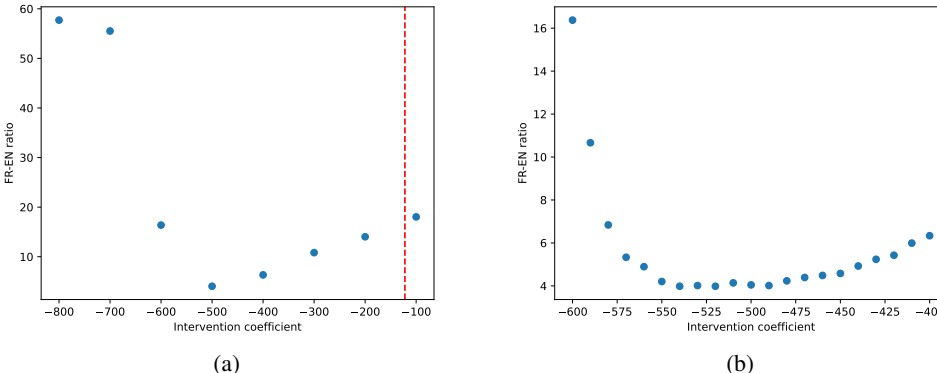

Figure 27: SAE FR-EN ratio (%) over different intervention strengths. We start from the recommended strength, -122, following Neuronpedia (https://www.neuronpedia.org/gemma-2-2b-it/steer). We determine the optimal intervention coefficient use a coarse search (a) followed by a fine-grained search (b). The best SAE result still underperforms our "multilingual" module's performance.

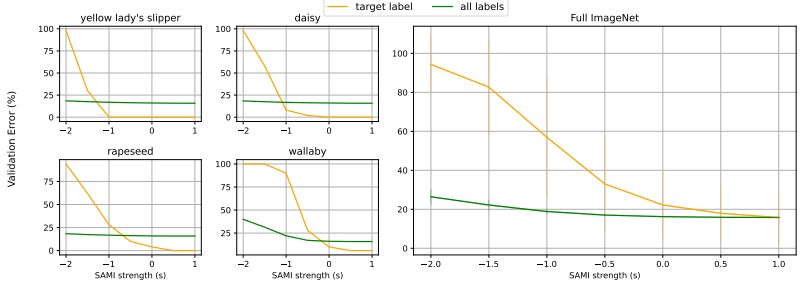

Figure 28: SAMI results on VIT-L/16. *Left:* 4 labels with the lowest/highest generalization error after intervention. *Right:* Average effect across all labels of ImageNet. With our discovered modules, under negative intervention, we disable the recognizability on the attacked target (orange curve). We also show the generalization error (green curve) on the full validation set.

s

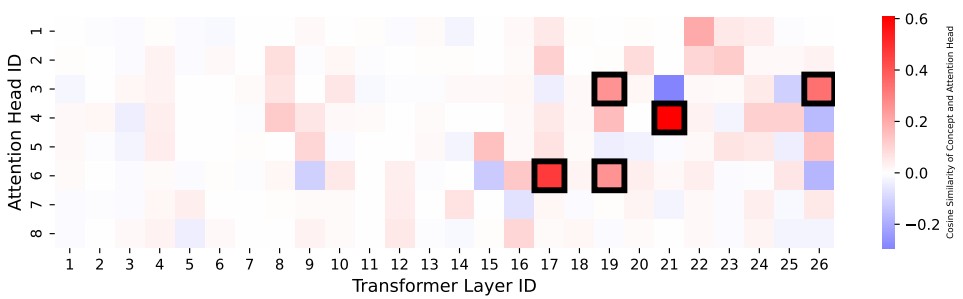

Figure 29: **"Dog" module:** We determine the "Dog" concept vector $v_c$, and compare its cosine similarity to every attention head use *half* of the original data. The black bounding boxes mark the 5 most important heads, and hence define the module.

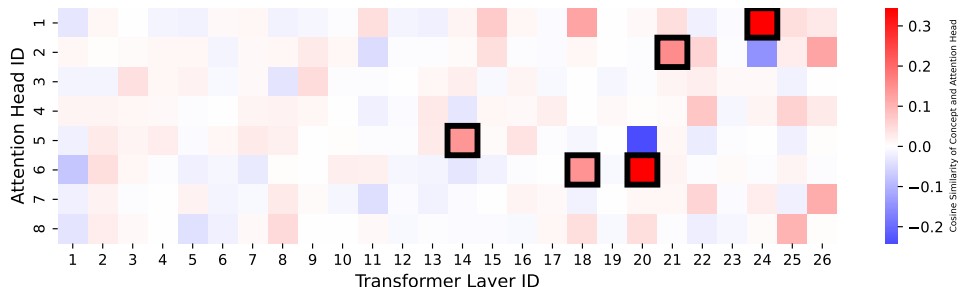

Figure 30: **"YELLING" module:** We determine the "YELLING" concept vector $v_c$, and compare its cosine similarity to every attention head use *half* of the original data. The black bounding boxes mark the 5 most important heads, and hence define the module.

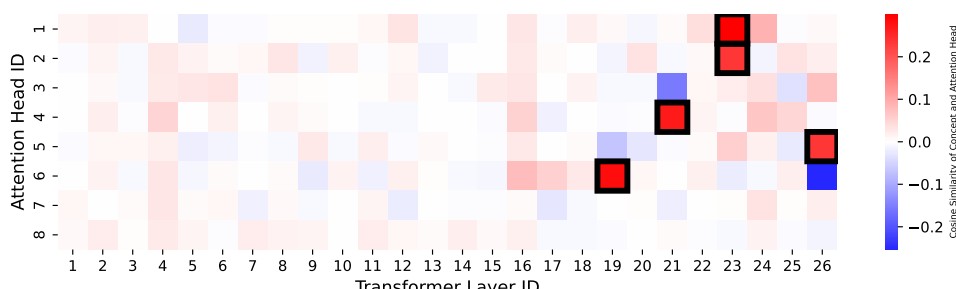

Figure 31: **"San Francisco" module:** We determine the "San Francisco" concept vector $v_c$, and compare its cosine similarity to every attention head use *half* of the original data. The black bounding boxes mark the 5 most important heads, and hence define the module.

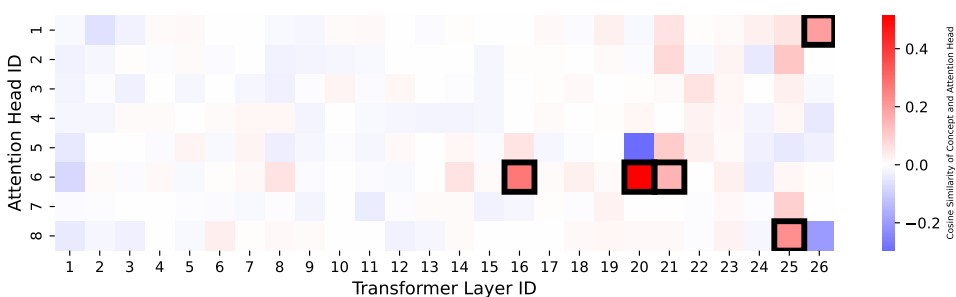

Figure 32: **"Multilingual" module:** We determine the "French" concept vector $v_c$, and compare its cosine similarity to every attention head use *half* of the original data. The black bounding boxes mark the 5 most important heads, and hence define the module.

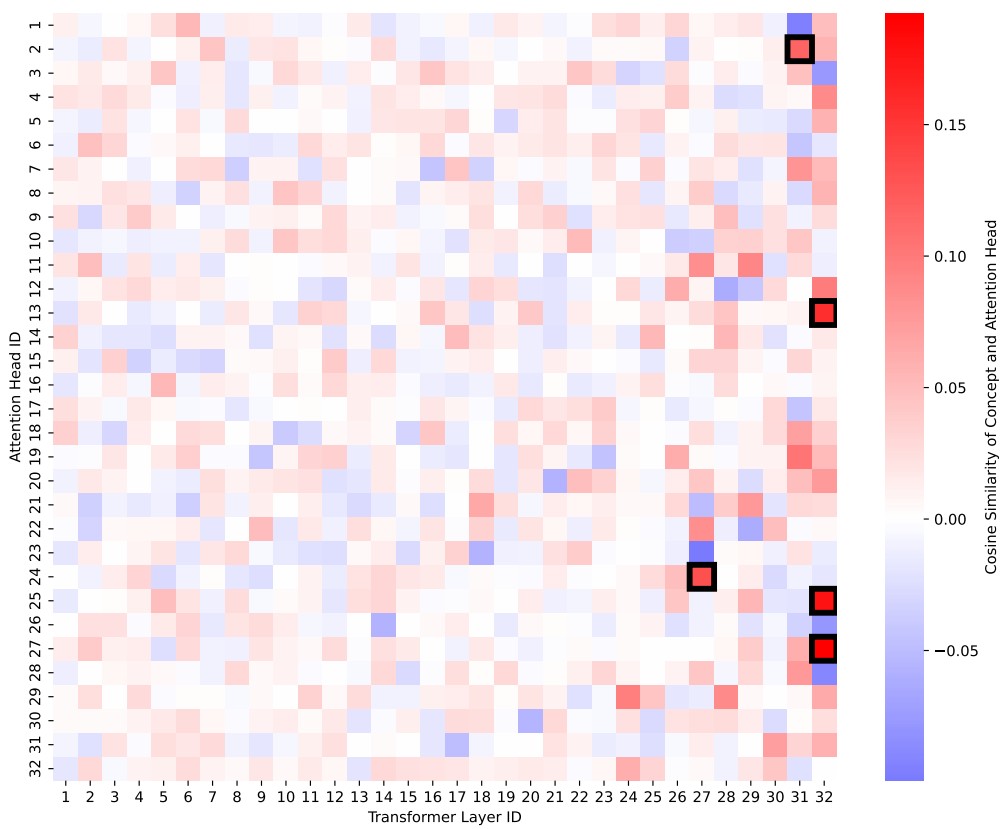

Figure 33: **CoT module:** We determine the CoT concept vector $v_c$, and compare its cosine similarity to every attention head in LLAMA3.1-8B-INSTRUCT use *half* of the original data. The black bounding boxes mark the 5 most important heads, and hence define the module.

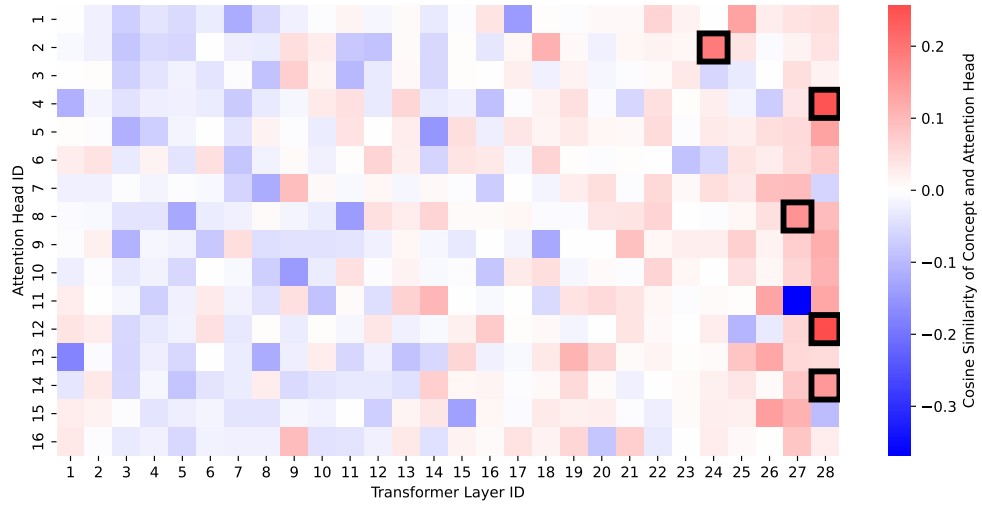

Figure 34: **CoT module:** We determine the CoT concept vector $v_c$, and compare its cosine similarity to every attention head in GEMMA-7B-BASE use *half* of the original data. The black bounding boxes mark the 5 most important heads, and hence define the module.

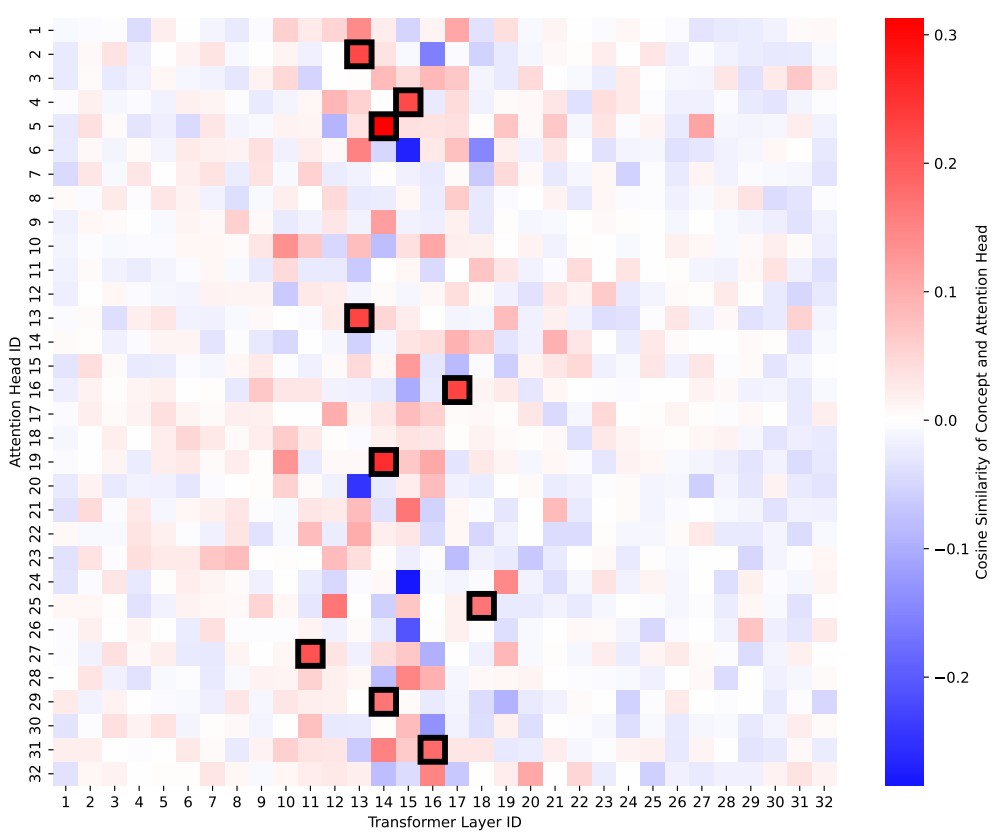

Figure 35: **Safety module:** We determine the safety concept vector $v_c$, and compare its cosine similarity to every attention head in LLAMA-2-CHAT-7B use *half* of the original data. The black bounding boxes mark the 5 most important heads, and hence define the module.

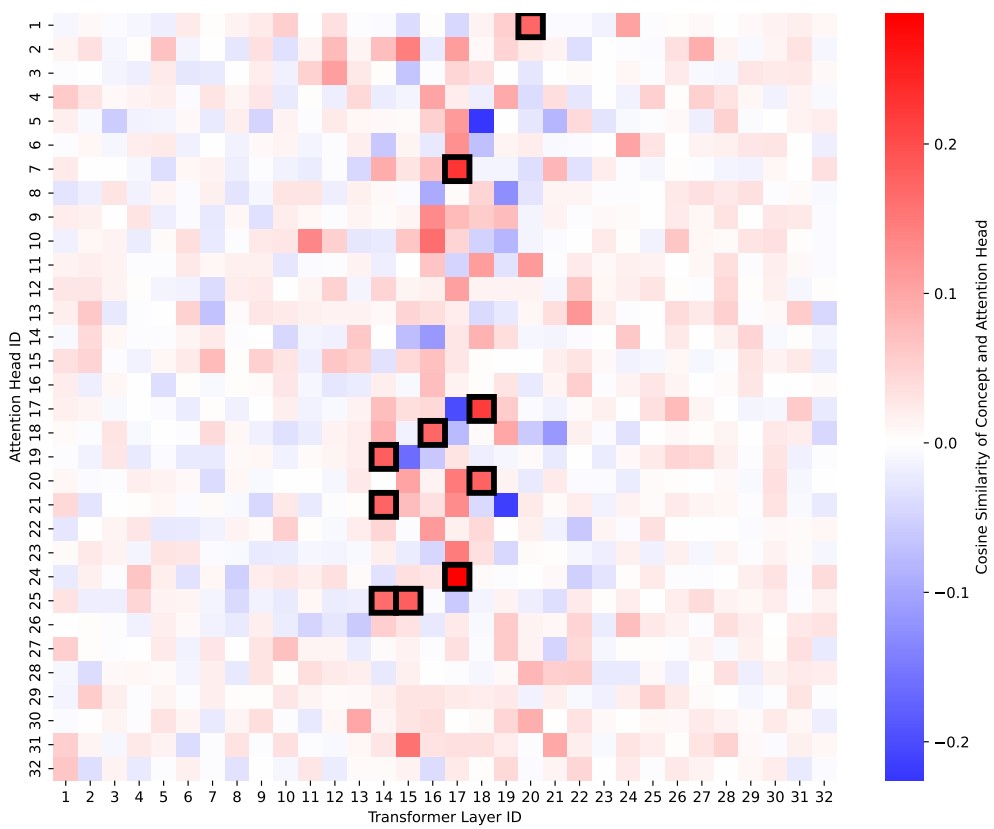

Figure 36: **Safety module:** We determine the safety concept vector $v_c$, and compare its cosine similarity to every attention head in QWEN-7B-CHAT use *half* of the original data. The black bounding boxes mark the 5 most important heads, and hence define the module.

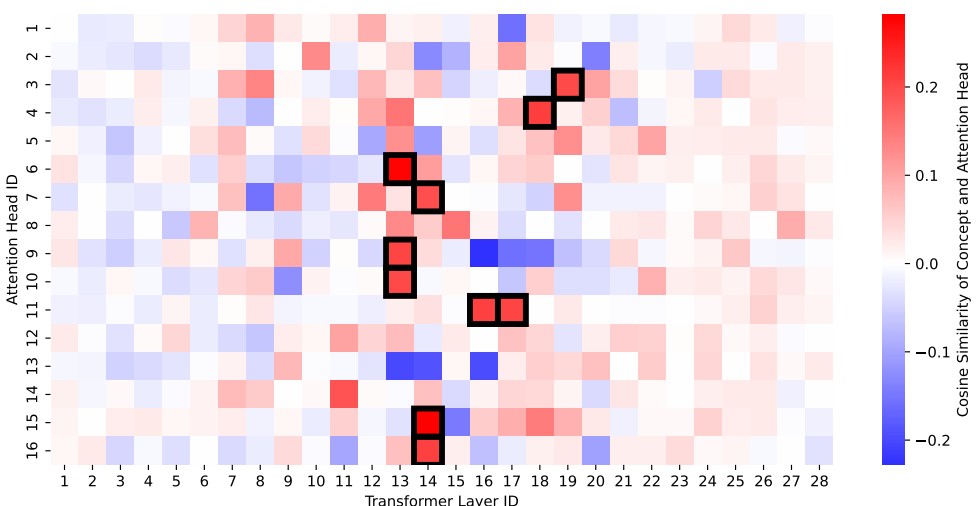

Figure 37: **Safety module:** We determine the safety concept vector $v_c$, and compare its cosine similarity to every attention head in GEMMA-7B-IT use *half* of the original data. The black bounding boxes mark the 5 most important heads, and hence define the module.

```
Prompt:   ¿Cuál es el idioma de este mensaje?
Generation (no intervention):   El idioma de este mensaje es español.
Generation (negative intervention):   This message is in English.
```

Figure 38: **"Multilingual" intervention on Spanish input:** the negative intervention compels the model to reply in English.

```
Prompt:              当前提示词输入的语言是什么?
Generation (no intervention):            当前提示词输入的语言是中文。
Generation (negative intervention):   The current prompt word input is in English.
```

Figure 39: **"Multilingual" intervention on Chinese input:** the negative intervention compels the model to reply in English.

```
Prompt:   In welcher Sprache ist diese Aufforderung formuliert?
Generation (no intervention):   Die Aufforderung ist in Deutsch formuliert.
Generation (negative intervention):   This prompt is written in English.
```

Figure 40: **"Multilingual" intervention on German input:** the negative intervention compels the model to reply in English.

```
Prompt:   ما هي لغة هذا الموجه؟
Generation (no intervention):   أنا نموذج لغوي كبير، تم تدريبي بواسطة جوجل.
Generation (negative intervention):   This is a text-based AI, so it doesn't have a "language" in the
traditional sense.  \n \n However, the language used to communicate with me is English.
```

Figure 41: **"Multilingual" intervention on Arabic input:** the negative intervention compels the model to reply in English.

