# OpenReview forum: "From Concepts to Components: Concept-Agnostic Attention Module Discovery in Transformers"
_ICLR.cc/2026/Conference — ICLR 2026 Poster_

### Official Review · Reviewer_WBy6 · 2025-10-23

**Soundness:** 3
**Presentation:** 2
**Contribution:** 2
**Rating:** 4
**Confidence:** 4

**Summary:**

This paper presents Scalable Attention Module Discovery (SAMD), a method for identifying attention heads associated with abstract “concepts” in transformers via cosine similarity, and Scalar Attention Module Intervention (SAMI), a one-parameter mechanism to amplify or suppress their influence. The approach is demonstrated across LLMs and ViTs, with qualitative interpretability results and modest quantitative effects on reasoning, safety, and vision benchmarks.

**Strengths:**

- Introducing attention-head–level concept attribution is an original direction that is computationally light and easily applicable to diverse transformer architectures.

 - The same pipeline is used across text and vision models, suggesting potential generality and extensibility.

 - The paper contributes to ongoing efforts to connect internal transformer components to semantic behaviors, particularly through sparse, interpretable “modules.”

**Weaknesses:**

- The evaluation is dominated by qualitative visualizations and anecdotal examples. There are no robust statistical analyses, reproducible metrics, or causal validation to confirm that the discovered modules truly mediate the claimed concepts.

 - The use of cosine similarity as a proxy for conceptual alignment is not theoretically or empirically justified; results may reflect correlation, not causation.

 - Choices of K (number of heads) and s (scaling factor) appear arbitrary, tuned via small grid searches without sensitivity analysis.

 - The paper never clearly defines in what sense the approach is “concept-agnostic.” This weakens the interpretive and theoretical clarity of the contribution.

**Questions:**

How is a “concept” defined in this framework, and how can the method be considered “concept-agnostic” if it depends on concept-specific vectors?

Can the identified attention modules be causally validated (e.g., via path patching or feature ablation)?

How robust are the discovered modules to randomization, different seeds, or model variants?

Could quantitative measures (e.g., mutual information or probing accuracy) strengthen the claims?

The heatmap figures are difficult to distinguish between colors, which are somehow important (e.g., with bold borders) and which are not.
Please, change the color.

---

> ### Author Response · Authors · 2025-11-18
> **Rebuttal by Authors (1/3)**
>
> Dear reviewer WBy6,
>
> We sincerely thank you for thoroughly reading our paper! Here are our responses to address all your concerns:
>
> *The evaluation is dominated by qualitative visualizations and anecdotal examples. There are no robust statistical analyses, reproducible metrics, or causal validation to confirm that the discovered modules truly mediate the claimed concepts.*
>
> We respectfully disagree with the judgment on our evaluation. We provide quantitative results in Section 4.1 to evaluate the effectiveness of our "French" module; in Section 4.2 to assess the intervention result of our "reasoning" modules; in Section 4.3 to demonstrate the superior jailbreak result by manipulating the "safety" modules; and in Section 4.4 to show the effect of intervention on recognition modules in ViTs. The qualitative visualizations are used to show the *location* of our discovered modules, and the anecdotal examples in Section 4.1 are collected on concepts that are nontrivial to evaluate quantitatively. For our quantitative evaluations, in Section 4.2, we use lm-eval-harness, a well-known repository to assess LLM capability with 10.6K+ stars; in Section 4.3, we follow the standard practice of HarmBench [1], a canonical benchmark designed to assess resilience against jailbreaking; in Section 4.4, we take the Vision Transformers on ImageNet-1K, a common model-dataset combination setup for vision research. For reproducibility, we plan to release the code upon acceptance. Finally, it is not suitable to launch *causal validation* experiments for our work, please see our response to your question *"Can the identified attention modules be causally validated"* below.
>
> [1] HarmBench: A Standardized Evaluation Framework for Automated Red Teaming and Robust Refusal
>
> *The use of cosine similarity as a proxy for conceptual alignment is not theoretically or empirically justified; results may reflect correlation, not causation.*
>
> Thank you for the question. We would be eager to argue that the use of the cosine score as a similarity measure in the vector space itself has a long-standing history with theoretical grounding and empirical support, as listed in the following references among related research works. For example, dating back to 2007, cosine similarity has been used to check the relatedness of texts in Explicit Semantic Analysis [2]. For theoretical justification, [3] illustrates that for all common word vectors, cosine similarity is essentially equivalent to the Pearson correlation coefficient. More recently, in the study of LLM representation, closeness is measured using cosine similarity [4]. Finally, we have explicitly acknowledged that our finding could reflect correlation in our limitation Section in Appendix F. We would emphasize that drawing *causal* conclusions is extremely hard: for example, [5] shows in previous MLP attribution works, even though the process is causality-inspired, attribution process does not inform good editing, and thus it questions the *causal* claims made in those references. We thus would want to argue: (1) the use of cosine similarity itself has its own justification through both theory and practical evidence, and we adopt it following existing work; and (2) the concern related to causation should not be treated as a weakness of our work, but a general view that the research community should be aware of.
>
>
> [2] Computing semantic relatedness using Wikipedia-based explicit semantic analysis
>
> [3] Correlation Coefficients and Semantic Textual Similarity
>
> [4] Scaling Monosemanticity: Extracting Interpretable Features from Claude 3 Sonnet
>
> [5] Does Localization Inform Editing? Surprising Differences in Causality-Based Localization vs. Knowledge Editing in Language Models
>
> *Choices of K (number of heads) and s (scaling factor) appear arbitrary, tuned via small grid searches without sensitivity analysis.*
>
> We respectfully disagree with this argument. The choice of $K$ (number of heads), as described in the paper and as shown in Appendix D, is determined by a stark separation of top cosine similarity values. For the choice of $s$, we have $s=1$ as the unmodified forward pass, and $s=-1$ as reversing the contribution of the discovered attention module. The grid search over $s$ is used to obtain the best result available based on these clear intuitions. We would also like to point out that another line of work, *steering vector-based intervention*, always require an extensive search over its intervention coefficient, as we have mentioned in Section 4.1, and as evidenced in *e.g.,* the Neuronpedia page (https://www.neuronpedia.org/gemma-2-2b-it/steer) or the Scaling Monosemanticity paper above [4]. Compared to this line of work, it is straightforward in our case to determine an initial intervention strength and do a small grid search to find the best parameter. Further, our results in Sections 4.1 and 4.3 show better performance compared to this baseline.

---

> > ### Author Response · Authors · 2025-11-18
> > **Rebuttal by Authors (2/3)**
> >
> > *The paper never clearly defines in what sense the approach is “concept-agnostic.” This weakens the interpretive and theoretical clarity of the contribution.*, and *How is a “concept” defined in this framework, and how can the method be considered “concept-agnostic” if it depends on concept-specific vectors?*
> >
> > Thank you for raising these questions. We first address your question related to concept definition. We have defined "concept" in Section 2.2. We acknowledge that the meaning of "concept" can vary across contexts, and it is generally elusive to define. Given the fact that a concept is generally thought of as representing a semantically meaningful unit, in this paper, we consider concepts to be **units that are operationalized by a positive dataset $D_p$, a collection of data points that all carry similar meaning or share a characteristic.** The shared characteristic is treated as the concept, and could be arbitrary as long as one could find a corresponding representative positive dataset.
> >
> > We proceed to the definition of "concept-agnostic" and apologize for any confusion it might bring. We use this term to contrast previous work, where human investigation and interpretation are strongly involved to obtain experimental results **on each independent concept studied** (see discussions in [6][7], and examples in [8][9]). Researchers have to try multiple independent methods with their manual inspection to decide which method to use in an ad-hoc fashion. There lacks a generic pipeline that can be adopted to broad concepts of interest. In contrast, our SAMD and SAMI pipeline is **agnostic** to the choice of concept, and we only rely on the concept vector abstraction and the corresponding dataset. Once obtained, our pipeline can be launched across multiple models, multiple modalities, across concepts that are either coarse (like reasoning and safety) or fine-grained (like specific nouns as "dogs"). We have made this statement clearer by adding a discussion in the introduction section to assist future readers.
> >
> >
> > [6] Towards Automated Circuit Discovery for Mechanistic Interpretability
> >
> > [7] Toward Transparent AI: A Survey on Interpreting the Inner Structures of Deep Neural Networks
> >
> > [8] Locating and editing factual associations in gpt
> >
> > [9] Interpretability in the Wild: a Circuit for Indirect Object Identification in GPT-2 small
> >
> >
> > *Can the identified attention modules be causally validated (e.g., via path patching or feature ablation)?*
> >
> > Thank you for proposing this question, and we would like to clarify that existing techniques, including causal interventions such as activation patching and feature ablation, and circuit-level interventions such as path patching, are not plausible for our method due to paradigm discrepancy. For the former, to effectively define a causal intervention, we have to first derive a **structural causal model (SCM)** over the internal variables of the transformers. After that, we could launch an intervention that replaces the structural equations of some variables with exogenously chosen values (*i.e.,* the do-operator). Existing causal interventions all rely on the **causal abstraction** where **internal representations are treated as variables, or nodes, while model weights themselves are fixed parameters that define the SCM, *not* variables inside it.** Since our intervention is directly applied on the attention module output weights, it is effectively changing this assumed SCM itself, not intervening in it. We could view model weights as the nodes of interest and construct a new SCM, and under this view, our intervention as well as our validation would already be treated as causal. However, this SCM is not the one commonly used in interpretability, and thus, we do not discuss this in our paper. For the latter, these circuit-level interventions can only operate on simple and synthetic concepts due to the strong reliance on the token probability measure (see *e.g.,* [9][10]). This is in contrast to the generic, complex concepts we have studied in our paper, which cannot be abstracted through simple token probability.
> >
> > [10] Localizing Model Behavior with Path Patching

---

> > > ### Author Response · Authors · 2025-11-18
> > > **Rebuttal by Authors (3/3)**
> > >
> > > *How robust are the discovered modules to randomization, different seeds, or model variants?*
> > >
> > > Thank you for this question. Given a model, our attention module discovery process is *deterministic* depending on the datasets $D_p$ and $D_n$, as described in Section 2.2. To address your concern, we have now conducted an experiment where we re-discover the modules using only **half** of the data compared to what we have done in our paper. We confirm that our proposed SAMD is pretty stable: the module remains largely unchanged under this situation. We have attached these visualizations to Appendix I. For the robustness across model variants, we believe our results in Sections 4.2, 4.3, and new results in Section 4.4 on ViT-L (please refer to our rebuttal to Reviewer BQcd) across multiple models and modalities have already highlighted the power and stability of our proposed pipeline.
> > >
> > >
> > > *Could quantitative measures (e.g., mutual information or probing accuracy) strengthen the claims?*
> > >
> > > Thank you for the question. To ensure we fully address your concern, could you please provide more detail on how mutual information is meant to be utilized in this context? We are aware of the fact that MI suffers from fundamental difficulties of exact estimation [11], and past work which relies on MI has shown different behaviors under different structures or estimators even on very simple neural networks (*e.g.,* [12][13][14]). For probing, could you please also elaborate on how it could be leveraged to strengthen our claims, since probing experiments are usually done on representations, while our pipeline uniquely focuses on attention module weights?
> > >
> > > [11] Formal Limitations on the Measurement of Mutual Information
> > >
> > > [12] Opening the black box of Deep Neural Networks via Information
> > >
> > > [13] On the Information Bottleneck Theory of Deep Learning
> > >
> > > [14] Understanding learning dynamics of binary neural networks via information bottleneck
> > >
> > >
> > >
> > > *The heatmap figures are difficult to distinguish between colors, which are somehow important (e.g., with bold borders) and which are not. Please, change the color.*
> > >
> > > Thank you for raising this concern. We will update the figures in a future version to better distinguish the attention heads with high cosine similarity scores.
> > >
> > > We hope our responses address your concerns and help you reassess the quality of our paper. Please notify us if you have further questions. We would be glad to discuss further!

---

> > > > ### Comment · Reviewer_WBy6 · 2025-11-24
> > > >
> > > > Thank you for running the additional experiment on subsampled data. It is helpful to see that the discovered modules remain largely unchanged when using only half of the dataset. This does increase my confidence that SAMD is at least stable with respect to dataset size.
> > > >
> > > > ------------------------
> > > >
> > > > Thank you for the response. To clarify, the question was not specifically requesting mutual information estimation or advocating for MI-based methods. I fully agree that MI estimation is challenging and may not be suitable here.
> > > >
> > > > The broader point is that some auxiliary quantitative measure,any measure beyond cosine similarity, could help validate that the identified heads meaningfully correspond to the intended concepts.
> > > >
> > > > For example, in representational interpretability, it is common to use:
> > > >  - probing classifiers to check whether a representation encodes a target feature, or
> > > >  - CAV-style evaluations (where high linear probe accuracy indicates that a concept vector actually captures the intended concept).
> > > >
> > > > My intention was simply that some analogous check could help strengthen the paper's claims.
> > > > Since SAMD relies entirely on cosine similarity as the identifying signal, an orthogonal quantitative measure, probing accuracy, linear separability, or even a simple diagnostic classifier, would provide independent evidence that the discovered "concept vectors" and "modules" are not artifacts of the chosen metric.
> > > >
> > > > This was meant as a general suggestion for broader validation, not a prescription to use MI specifically.
> > > >
> > > > ------------------------
> > > >
> > > > Thank you for acknowledging the issue. To clarify the concern: the problem is not only the general color palette but that, in several figures, different attention heads appear with identical or nearly identical colors, making the heatmaps impossible to interpret reliably.
> > > >
> > > > For example, in Fig. 11 ("yelling") line mentioned before, attention heads at layer 18, head 1 and layer 18, head 6 display the exact same color value, while one has a black border.

---

> > > ### Comment · Reviewer_WBy6 · 2025-11-24
> > >
> > > The rebuttal explains that “concept-agnostic” means the method does not rely on per-concept manual inspection or bespoke methods (unlike prior work).
> > > This framing does not appear in the introduction or method section. In the submitted paper, “Concept-agnostic” is asserted, but not explained.
> > >
> > > --------------------------------
> > >
> > > Thank you for the detailed explanation. I agree that full causal modeling of weight-level interventions is outside the scope of this paper. However, the rebuttal does not really address the original intent of the question.
> > >
> > > The question was not requesting a full SCM or path-patching-style causal analysis. It simply pointed out that the paper makes claims about “modules mediating concepts” without providing even lightweight validation (e.g., sanity checks, alternative metrics, or simple ablations) that the identified heads meaningfully influence model behavior.
> > >
> > > Since you now clarify that:
> > > - causal tools used in activation-level studies cannot be easily applied here, and
> > > - causal validation is not a goal of the method.
> > >
> > > I agree that this issue can be considered out of scope.

---

> > > > ### Author Response · Authors · 2025-11-27
> > > > **Rebuttal by Authors**
> > > >
> > > > *Validity of Cosine Similarity*
> > > >
> > > > We respectfully and strongly disagree with these statements. The reviewer stated, “the use of cosine similarity is not justified; results may reflect correlation, not causation” in the original review, while also stating, “the issue is not causality” in the response, which forms a direct contradiction. We have already provided references to the widely accepted use of the cosine score as a similarity measure and have acknowledged the possible limitation of correlation in the limitations section. We will add these references to the revised draft. The reviewer insisted that we launch experiments using alternative measures. However, we argue that whether such measures (like L2 distance) could work for module attribution is beyond the scope of our paper, since our focus is not on conducting an exhaustive search over all possible quantitative measures to find the best one. Finally, could the reviewer clarify what is meant by “in aggregate” in the comment: “whether cosine similarity correlates with measurable behavioral influence even in aggregate”?
> > > >
> > > > *Explanation of Concept-Agnostic*
> > > >
> > > > The reviewer states, "this framing does not appear in the introduction or method section," in the response. However, as we stated in the rebuttal, we have **already** added the related discussion in the introduction in the revised draft.
> > > >
> > > > *Causal modeling*
> > > >
> > > > We are confused about the statements made by the reviewer. The reviewer first states “the rebuttal does not really address the original intent of the question”, and concludes with “I agree that this issue can be considered out of scope”. Could the reviewer clarify and assert that this question has been addressed?
> > > >
> > > > *Quantitative measure to strengthen the claims*
> > > >
> > > > The reviewer suggested using mutual information or probing accuracy to strengthen our claims in the original review. As we stated in the rebuttal and as agreed by the reviewer, mutual information is challenging and not suitable in this context. Also, as we stated in the rebuttal, representation probing is not suitable because we are operating on attention module weights. However, **all** new suggestions from the reviewer in the response, including *probing accuracy, linear separability, or even a simple diagnostic classifier,* are still examples on representations which cannot be applied to our modules. Thus, we argue that these suggestions are out of the scope of our paper.

---

> > ### Comment · Reviewer_HhFz · 2025-11-21
> >
> > Regarding the following questions from the reviewer:
> >
> > > The evaluation is dominated by qualitative visualizations and anecdotal examples. There are no robust statistical analyses, reproducible metrics, or causal validation to confirm that the discovered modules truly mediate the claimed concepts.
> >
> > > The use of cosine similarity as a proxy for conceptual alignment is not theoretically or empirically justified; results may reflect correlation, not causation.
> >
> > > Choices of K (number of heads) and s (scaling factor) appear arbitrary, tuned via small grid searches without sensitivity analysis.
> >
> > I, like the authors, also respectfully disagree with these comments and find their responses to these to be strong and compelling.

---

> ### Comment · Reviewer_WBy6 · 2025-11-24
>
> Thank you for the rebuttal. While I appreciate the clarification, the core concern about quantitative rigor remains unresolved.
> ***Selective and Incomplete Evaluation***
> Section 4.1 evaluates only four SAE concepts (“dog,” “San Francisco,” “yelling,” “French”) despite many more available in Neuronpedia.
> For multilingual analysis, omitting “British English” or other languages makes the “French module” claim much less compelling. The small and selective set of concepts raises concerns of cherry-picking.
> ***Presentation Issues Undermine Confidence***
> For example, in Fig. 11 (“yelling”), two distinct heads (layer 18, heads 1 and 6) have identical color values, making interpretation ambiguous. Many appendix figures suffer similar clarity issues, which weakens trust in module identification.
> ***Standard Benchmarks ≠ Rigorous Evaluation***
> Using lm-eval-harness, HarmBench, and ImageNet does not address the actual criticism.
> The issue is not the datasets but the evaluation design, which still lacks:
>  - statistical robustness (no repeated runs, no variance or uncertainty),
>  - systematic coverage (only a few concepts receive quantitative analysis),
>  - ablations (no sensitivity to K or s, or even some discussion of s for labels based on Fig. 9).
>
> These gaps remain unaddressed in the rebuttal.
>
> --------------------------------
>
> Thank you for the explanation. However, the rebuttal does not fully address the core issue.
> The concern is not whether cosine similarity has been used in prior NLP research, it certainly has, but whether in this paper it is justified as an appropriate metric for identifying concept-bearing attention heads.
> ***References introduced in the rebuttal are not in the paper***
> Some references used to defend cosine similarity were not cited or discussed in the original submission.
> This means the justification was not originally part of the method nor its stated assumptions.
> ***The issue is not causality***
> As stated earlier, I am not insisting on causal guarantees.
> The concern is simply that:
>  - cosine similarity is assumed to be meaningful in this context,
>  - yet no empirical sanity checks or ablations validate this assumption.
> For example, the paper does not show:
>  - whether alternative metrics (L2 distance, centered cosine, projection magnitude) produce different or identical modules,
>  - whether cosine similarity correlates with measurable behavioral influence even in aggregate.
> Without such checks, the numeric scores may be valid or may simply reflect incidental correlations.
>
> --------------------------------
>
> Thank you for the clarification. However, the concerns about the arbitrariness of K and s remain unresolved.
> ***The claimed “stark separation” for K is not consistently visible***
> While the rebuttal states that K is chosen based on “stark separations,” the figures in Appendix D (Figs. 22–25) do not consistently show such clear thresholds.
> In some cases, the cosine similarities:
>  - overlap visually,
>  - decline gradually,
>  - or show several near-ties within the top 5.
> Given the presentation issues (overlapping colors, ambiguous scaling), it is difficult to objectively verify that K = 5 always emerges naturally.
>
> Thus, the choice of K still appears heuristic rather than derived from a principled or quantitative rule.
>
> ***Grid search over s is still ad hoc without sensitivity analysis***
>
> Even if s = 1 (none) and s = –1 (“reversal”) offer intuitive anchors, the paper still tunes s through a small grid search without:
>  - reporting the search range,
>  - showing sensitivity of results to different s values,
>  - verifying that the discovered heads respond consistently across strengths,
>  - or checking whether the same s generalizes across concepts or datasets.
> This makes the intervention strength feel opportunistic rather than robustly justified.
>
> ***Presentation issues amplify the uncertainty***
> Because several Appendix D plots are difficult to read (color overlaps, thin separations, inconsistent scaling), the reader cannot reliably verify the claimed separation of top-K heads. This affects confidence in K-selection especially.
>
> --------------------------------
>
> Across the last three issues, the core concern is that the paper lacks quantitative rigor and methodological robustness, and the authors’ rebuttal does not adequately address these points.

---

> > ### Author Response · Authors · 2025-11-27
> > **Rebuttal by Authors**
> >
> > Dear Reviewer WBy6,
> >
> > Thank you for your comments. Here are our responses to your concerns.
> >
> > *Selective and Incomplete Evaluation*
> >
> > We respectfully and strongly disagree with the statement, especially on the claims “despite many more available in Neuronpedia”, “British English”, and “cherry-picking”. As we have **already** explained in Section 4.1, Appendix B.2, and in the rebuttal, **for our study, we use the only publicly available set of SAE features with accompanying interpretations, and exclude abstract concepts that are difficult to evaluate even qualitatively (e.g., “bravery”, “humor”).** To be more precise, the available SAE features listed on the neuronpedia page include:
> > - Dog
> > - French
> > - Yelling
> > - Bravery
> > - British English
> > - Misspelling
> > - Spanish
> > - San Francisco
> > - Humor
> > - Cringe
> >
> > As one can clearly see, “bravery”, “British English”, “misspelling”, “humor”, and “cringe” are concepts that are hard to measure, even qualitatively. Specifically, the reviewer suggested using “British English” for multilingual analysis, which does not fit the context and is incorrect because **British English is not a language distinct from English**. That said, we have launched new experiments using the Spanish feature. We are excited to see that this feature leads to the Spanish module which shares **exactly the same set of attention heads that forms our French module.** Further tests reveal that this module is in fact responsible for the broader multilingualism capability of the LLM: the intervention works not only on French or Spanish inputs but also on other languages such as Mandarin Chinese, Hindi, and Arabic. We are updating our revised draft to reflect this new result, and we believe these additional observations help strengthen our paper. We want to emphasize that this intrinsic property of attention modules cannot be easily obtained by observing only the feature space, as done in previous work, and thus highlights the unique advantage of our methodology.
> >
> > *Comments on Figure Presentations, selection of K*
> >
> > We agree on the original suggestion that the heatmap plots could be improved by finding a different color set, and we are working on it. However, we respectfully and strongly disagree with the comment on “identical color values”, “ambiguous interpretation”, “inconsistent scaling”, “ambiguous scaling”.
> > - Color value and interpretation. If the color values for different heads are close, that is simply because their cosine similarity values are not far away. The plot is handled by matplotlib, not by us. In Section 4.1, we select $K=5$ as a *consistent* hyperparameter across all 4 concepts we analyzed, because (1) it is the number that the attention head cosine similarity scores across all concepts show a separation, and (2) we want to highlight the simplicity of our proposal to quickly find attention modules for target concepts, and thus we do not apply a per-concept sweep over $K$. Further, we never claim that *“$K=5$ always emerges naturally.”* In practice for the best performance, one should measure the scores and select the module based on their judgment, and we have already acknowledged this in our limitation section that **“we do not emphasize causality for the module we identify through SAMD”** and **“our attention module could be either overcomplete or incomplete.”** Even for the “yelling” module the separation is not as strong as the rest concepts, we still observe positive results.
> > - “Scaling”. Could the reviewer please explain what they mean by inconsistent scaling or ambiguous scaling? **We do not use any scaling in our heatmap plots or cosine score plots**, and thus the statement is not factual. We are very curious about how the reviewer derives such a conclusion and uses this as evidence to criticize our visualization.
> >
> > *Standard Benchmarks ≠ Rigorous Evaluation*
> >
> > We respectfully and strongly disagree with the statement. We have to reiterate and restate that we are not only using **community-standard benchmarks**, but also following **community-standard evaluation protocols.** Repeated runs, variance or uncertainty are not considered because the evaluation protocol uses greedy decoding, and we follow the standard usage of these benchmarks. Regarding systematic coverage, we have already explained in our original rebuttal that some concepts are not suitable for quantitative evaluations. Regarding discussion of $s$ for labels based on Figure 9, we have already stated in the caption that **we choose 4 labels with the lowest/highest generalization error after intervention** and observe consistent behavior. We have already explained the selection of $K$ and $s$ in the original rebuttal, and stated that the search over $s$ is only used to obtain the best result around the initial values of $1$ and $-1$.

---

### Official Review · Reviewer_HhFz · 2025-10-27

**Soundness:** 4
**Presentation:** 4
**Contribution:** 4
**Rating:** 8
**Confidence:** 4

**Summary:**

This paper introduces an attribution method (of model components, not the input) for explaining transformer-based models. Specifically, it identifies the most important attention heads across the model which are relevant to a concept of interest i.e. the French language. To do so, they identify the maximally close (in terms of cosine similarity) attention heads to concept vectors captured from a positive dataset. They then propose that the identified components of the model can be up- or downscaled to change model behavior, thus verifying their component extraction, and providing a real use case for the method.

**Strengths:**

Very interesting method with good novelty. Unlike many previous MLP neuron attribution approaches, this is the first I have seen which identifies attention concepts. This is an interesting and critical result with the stronger push for mechanistic interpretability and adjacent approaches in the modern XAI literature.

Well written with extensive experimental results.

I think the simplicity of the approach is a benefit to its usability. I feel that I could replicate this with a few hours of work if I had access to the datasets.

**Weaknesses:**

Minor – plainly calling this an attribution method feels misaligned with the literature. Attribution methods often refer to input (feature) attribution. This is more aligned with neuron attribution. Perhaps it should be attention attribution but not to be confused with input attribution using attention weights/gradients.

There are not any true comparisons against other methods. It is hard to tell if this should be negative because it may be challenging to create a fair comparison against a similar MLP based method. I think it could have been possible to use knowledge editing benchmark.

**Questions:**

Did the authors consider a knowledge editing style benchmark?

---

> ### Author Response · Authors · 2025-11-18
> **Rebuttal by Authors**
>
> Dear reviewer HhFz,
>
> We sincerely thank you for recognizing the novelty and strength of our paper! Here are our responses to address all your concerns:
>
> *Minor – plainly calling this an attribution method feels misaligned with the literature. Attribution methods often refer to input (feature) attribution. This is more aligned with neuron attribution. Perhaps it should be attention attribution but not to be confused with input attribution using attention weights/gradients.*
>
> Thank you for highlighting this. We have explicitly stated this crucial difference in the revised abstract to avoid confusion for future readers.
>
> *There are not any true comparisons against other methods. It is hard to tell if this should be negative because it may be challenging to create a fair comparison against a similar MLP based method.*
>
> Thank you for raising this concern, and yes, it is indeed challenging to compare to MLP-based methods! Our pipeline consists of two steps: the attention module discovery step and the intervention step. For the module discovery step, as we reviewed in Related Work Section B.1, there are no directly comparable methods to attribute model behavior to attention modules. For the intervention step, we are the first and only work to intervene on model components and observe the change in outputs, and thus, there are also no directly comparable baselines along this line. These are the reasons we choose steering vector-based intervention as baselines in Sections 4.1 and 4.3. For MLP-based methods, we refer to our Related Work Section B.1 for more details. Existing work on that direction cannot conduct attribution for arbitrary concepts as we do; instead those methods extensively focus on factual associations (knowledge that consists of a subject and an object with a certain relation, for example, Paris is located in *France*). This makes a direct comparison to those methods highly nontrivial. Moreover, in previous MLP-based studies, even for factual associations, successful attribution does **not** inform good **editing** as shown empirically in [1]. This makes a stark contrast to our study, where our intervention clearly leads to the anticipated effect one would expect.
>
> [1] Does Localization Inform Editing? Surprising Differences in Causality-Based Localization vs. Knowledge Editing in Language Models
>
> *Did the authors consider a knowledge editing style benchmark?*
>
> Thank you for bringing this into the discussion! This is a fantastic suggestion; however, based on current knowledge editing benchmarks that we are aware of (for example, a well-recognized benchmark, KnowEdit [2]), factual association-style prompts dominate. The "knowledge", or "concept", considered in those benchmarks is limited to nouns defined through a single prompt, and lacks an associated dataset. Given that our attribution and intervention strongly rely on an arbitrary concept **and the associated dataset** used to construct the concept vector, it is hard to apply our methods directly to such benchmarks.
>
> [2] A Comprehensive Study of Knowledge Editing for Large Language Models
>
> We hope our responses address your concerns. Please let us know if you need further clarification, we would be delighted to discuss further!

---

> > ### Comment · Reviewer_HhFz · 2025-11-21
> >
> > Thank you for the well-crafted responses.
> >
> > You make a great counterpoint regarding my knowledge editing question. Perhaps the applications of this work would be more suited towards machine unlearning. I am not necessarily asking for such an evaluation, but I am curious what the thoughts of the authors' are on this, either in terms of the downstream uses of their method, or the applicability of such a benchmark for evaluation of their method.

---

> > > ### Author Response · Authors · 2025-11-27
> > > **Rebuttal by Authors**
> > >
> > > Dear Reviewer HhFz,
> > >
> > > We sincerely thank you and welcome your suggestions to improve our work! We understand your intent and are glad to share our thoughts with you. We do not place our work on the same level as machine unlearning because we believe there is a crucial difference: unlearning targets the removal of certain information or capabilities **without affecting the rest of the model.** However, in our module discovery and intervention, there can be overlaps between different modules. For example, the fourth head in layer 21 of Gemma-2-2B-it is included in both our "dog" module and our "San Francisco" module. Thus, intervening in one of them will likely affect the model's behavior with respect to the other. This is why we frame our pipeline as "discovery and intervention", and avoid using the terms "learning" or "unlearning" to prevent overclaiming.

---

> > > > ### Comment · Reviewer_HhFz · 2025-11-27
> > > >
> > > > Thank you for the discussion. This makes sense, especially in the context of the superposition hypothesis.
> > > >
> > > > Great work overall, thank you.

---

### Official Review · Reviewer_BQcd · 2025-10-31

**Soundness:** 3
**Presentation:** 3
**Contribution:** 3
**Rating:** 4
**Confidence:** 4

**Summary:**

The authors propose SAMD and SAMI - Scalable Attention Module Discovery and Intervention. Given a feature vector $v_c$, SAMD discovers a set of $K$ attention heads whose output $a_{l,h}$ has on average high cosine similarity to $v_c$. They call this set of attention heads (which is a circuit in a way) a module.  Given a the circuit of $K$ attention heads, SAMI amplifies or suppresses the module by scaling the attention heads output $a_{l,h}$ by a scaling factor $s$ which they choose on a per problem basis using grid search. They demonstrate the effectiveness of their methods in 4 different experiments:
a) They find and steer modules that correspond to SAE features. This motivates the name - as they choose concepts, and search for relevant modules related to the concept. This is done by generating a dataset $D_p$ for which an SAE feature $v_c$ is highly activated and then running SAMD with $v_c$ and $D_p$.
b) They find and steer a module that corresponds to reasoning capabilities in the model. They report improved scores on the GSM8K reasoning benchmark.
c) They find and steer a module that corresponds to a refusal direction. The report improved attack success rate over orthogonalization of the refusal direction in two out of three cases.
d) They find and steer a module that corresponds the classification of a given ImageNet target on ViT-B/32 21k, showcasing effectiveness of targeted unlearning of a single class.

**Strengths:**

* Novel and elegant method for circuit discovery
* Clear presentation of the findings
* Demonstration of effectiveness of method on a broad variety of applications over two different modalities. Especially in the vision literature this is addressing a research gap, as vision-circuit discovery remains under-explored.

**Weaknesses:**

* 4.2 the construction of $D_p$ is unclear from just reading the main body of the paper.
* Concept figure should be improved
	* Font way to small
	* No order of panels provided
	* SAMI is not explained in the rightmost panel
* Comparison to baseline such as e.g. difference in means is missing for 4.1, 4.2 and 4.4. If this concern is addressed appropriately I will improve my score.
* In 4.2 the authors only report evals on the dataset that they used for construction. An OOD reasoning benchmark eval would be useful to evaluate the generality of the reasoning-module.
* Only ViT-B 32 evaluated in 4.4). Experiments with at least ViT-L would be recommended as ViT-B often shows different behaviors from its bigger counterparts. If this concern is addressed I will improve my score.

**Questions:**

* Did the authors explore including highly negative cosine sim attention heads (e.g. Fig 24.) and flipping the $s$ value for these heads? If so, that did they find? Especially in Fig 24 gemma 7b one head seems to have the highest absolute alignment with -0.4 similarity while the highest positive alignment is only 0.3.
* What is the rational behind the $D_p$ construction via the test samples of GSM8K? Is it that the prompts are explicitly encouraging to reason?

---

> ### Author Response · Authors · 2025-11-18
> **Rebuttal by Authors (1/2)**
>
> Dear reviewer BQcd,
>
> We sincerely thank you for taking the time to read our work in great detail! Here are our responses to address all your concerns:
>
> *4.2 the construction of $D_p$ is unclear from just reading the main body of the paper.*
>
> We apologize for the confusion, and we would be eager to explain the construction of $D_p$ and the concept vector. In Section 4.2: reasoning module, the positive dataset $D_p$ is taken as the first 100 test prompts (without accessing the label information) in GSM8K constructed by lm-eval-harness, and the negative dataset $D_n$ is absent. All these prompts contain few-shot demonstrations to encourage Chain-of-Thought reasoning. We obtain the concept vector through **difference-in-means** as described in Equation (2) in Section 2.2 with the last layer representation. We have updated the text in Section 4.2 to more precisely reflect this construction.
>
> *Concept figure should be improved: Font way to small; No order of panels provided; SAMI is not explained in the rightmost panel*
>
> Thank you for these suggestions, and we will improve our concept figure following your advice to make it more friendly to future readers.
>
> *Comparison to baseline such as e.g. difference in means is missing for 4.1, 4.2 and 4.4. If this concern is addressed appropriately I will improve my score.*
>
> We believe there is a conceptual misunderstanding here, and we are happy to address your question. We want to clarify that **difference-in-means** is not a baseline intervention method, but instead a way to construct the concept vector, as described in Section 2.2. For baselines, our pipeline consists of two steps: the attention module discovery step and the intervention step. For the module discovery step, as we reviewed in Related Work Section B.1, there are no directly comparable methods to attribute model behavior to attention modules. For the intervention step, we are the first and only work to intervene on model components and observe the change in outputs, and thus, there are also no directly comparable baselines along this line. There are existing works to intervene on model *representations* using *steering vectors*, and to establish a comparison to plausible baselines as much as possible, we have adopted this baseline in: 1) Section 4.1 to quantify the effectiveness of altering model response language; and 2) Section 4.3 to assess the change with respect to model safety. To our knowledge, there are **no steering vectors available** for LLM reasoning or ViTs. In summary, we hope this could help clarify your question and explain that there are no missing baselines, but instead, we have tried our best to incorporate *steering vector-based intervention* in our paper to compare quantitatively.
>
> *In 4.2 the authors only report evals on the dataset that they used for construction. An OOD reasoning benchmark eval would be useful to evaluate the generality of the reasoning-module.*
>
> Thank you for the suggestion. Following your advice, we have conducted additional evaluations as follows. Use the "reasoning module" discovered with GSM8K, we have now added the MATH reasoning benchmark to evaluate out-of-distribution performance. With Llama-3.1-8B-Instruct, SAMI improved test accuracy from 39.78 to 40.58, and on Gemma-7B-Base, SAMI improved test accuracy from 24.16 to 24.74. We see that our intervention even carries over out-of-distribution, which is an important insight. Thank you for pointing this out, as we feel this further strengthens our claims. We have attached these new results to the updated paper.
>
> *Only ViT-B 32 evaluated in 4.4). Experiments with at least ViT-L would be recommended as ViT-B often shows different behaviors from its bigger counterparts.*
>
> Thank you for the suggestion. Following your recommendation, we have now conducted experiments with ViT-L and find no significant difference from our ViT-B baseline. This demonstrates the scalability of our pipeline to larger vision models. We attached the new results in Appendix G.

---

> > ### Author Response · Authors · 2025-11-18
> > **Rebuttal by Authors (2/2)**
> >
> > *Did the authors explore including highly negative cosine sim attention heads (e.g. Fig 24.) and flipping the value for these heads? If so, that did they find? Especially in Fig 24 gemma 7b one head seems to have the highest absolute alignment with -0.4 similarity while the highest positive alignment is only 0.3.*
> >
> > Thank you for raising this intriguing question. We did not explore this suite of experiments in our submitted version, because in the context of the concept vector we have discussed, **the opposite direction does not reliably correspond to the opposite meaning of a certain concept**. This suggests that a high absolute value with a negative score does not reliably indicate the importance of that head with a "negative concept". To address your concern and to validate this hypothesis rigorously, we explore whether intervening on those highly negative cosine similarity attention heads could lead to a similar phenomenon with respect to reasoning and safety (as in our Sections 4.2 and 4.3). Identical to the setup in these two sections, we take the Top-5 or Top-10 heads to formulate our module, but with the *negative* ones instead.
> >
> > The results corroborate our intuition. For reasoning, this intervention does not bring any improvement; on Llama-3.1-8B-Instruct, the intervention leads to lower accuracy than the baseline, and moreover, it brings Gemma-7B-Base accuracy to **0%**, likely because one attention head in the first layer is included. For safety, this intervention does not alter model behavior, as the jailbreak rate is not changed across all models no matter whether we choose to intervene positively or negatively. We have included the results in the updated Appendix H.
> >
> > *What is the rational behind the construction via the test samples of GSM8K? Is it that the prompts are explicitly encouraging to reason?*
> >
> > Yes! According to lm-eval-harness, these prompts are constructed with few-shot demonstrations to explicitly encourage Chain-of-Thought reasoning. The reason we select a subset of test samples is two-fold. First we aim to mimic the practice in test-time adaptation, where a model has access to test samples but not labels, to improve its own performance. Second, the GSM8K training set is commonly used in model training, and we want to avoid any risk that the concept vector is biased by training artifacts, overfitting patterns, or idiosyncrasies introduced during the training process.
> >
> > We hope our responses address your concerns and help you reassess the quality of our paper. Please do not hesitate to let us know if you need further clarification, we would be happy to discuss more!

---

### Author Response · Authors · 2025-11-18
**General Response by Authors**

We sincerely thank all reviewers for taking the time to review our paper and providing valuable feedback. We appreciate all reviewers for the consistent recognition of the novelty and elegance of our proposed pipeline, and the broad experimental coverage across both language and vision fields. Our key insight of the concept vector abstraction and the cosine similarity importance assignment allows us to discover attention modules across concepts with varying granularities, from simple nouns as “dogs” to highly abstract ones as “safety”. Beyond attention module discovery, our proposed intervention effectively diminishes or amplifies the target concept in both language and vision models, which is the first model-component-based intervention method that achieves such effects.

The reviewers have raised several important common concerns that we are eager to address.

1. The comparison to baselines and the use of qualitative demonstrations.

2. The robustness of our Scalable Attention Module Discovery.



We appreciate these insightful comments and are pleased to provide detailed clarifications to each of these points below.



**Comprehensiveness of our evaluation.** In our paper, we consider both the language and vision modality, demonstrating the generality of our proposed pipeline. Since we are the first to discover attention modules in general transformers and apply interventions on them, there is no directly comparable work. To facilitate a comprehensive evaluation, we compare our intervention against steering-vector-based baselines when applicable, in Section 4.1 and Section 4.3. The qualitative visualizations are used to show the location of our discovered modules, and we only provide qualitative demonstrations for the concepts that are not easy to measure quantitatively.



**Robustness of our Scalable Attention Module Discovery.** In our rebuttal, we claim the robustness of our method through the following three new sets of experiments.

1. Re-discover the modules using only half of the original data.  We find the modules remain largely unchanged under this situation, with 0-1 attention heads being different. We have attached these results in Appendix I.
2. Out-of-distribution test of the “reasoning module”. Using the module discovered with GSM8K, we demonstrate improved accuracy on the MATH benchmark, showing our intervention even carries over out-of-distribution. We have attached this result to our paper.
3. Vision experiment on larger ViTs. We launch our pipeline on a larger Vision Transformer, ViT-L/16, and obtain similar observations as we have seen on ViT-B/32. We have attached these results in Appendix G.



We would be eager to summarize our contributions and emphasize the importance of our work again as follows.


**The first attention module discovery method for arbitrary concepts on general transformers.** We propose a novel method that utilizes vector abstraction of concepts and finds important attention heads through cosine similarity. It addresses challenges such as the difficulty of representing concepts through concrete tokens and shows the importance of attention mechanisms for model attribution research.



**The first model component-based intervention method to steer model behavior.** Our proposed intervention only relies on a single scalar and is easily applicable to general transformer structures. Its effectiveness has been verified both quantitatively and qualitatively on diverse model families in both language and vision domains.



We sincerely thank all reviewers for their time on our rebuttal. We hope our response addresses your concerns and highlights the significance of our contributions. We would be delighted to discuss further in the rebuttal period.

---

### Meta-Review · Area_Chair_QQ1m · 2026-01-02

**Summary:**

The paper proposes a novel method for circuit discovery utilizing a multi-head attention module. The reviewers acknowledge the paper's value and novelty in circuit discovery and XAI, as well as its simplicity and extensibility. However, the reviewer also raised concerns about the lack of statistical analysis and comparison with other methods.

Two reviewers might rate acceptance scores [**Bqcd**, **HhFz**], while **WBy6** would stay in its negative score.
I carefully reviewed the discussion and found Reviewer WBy6's concerns to be valid and reasonable.
I didn't find a reason to ignore or downweight the review.

I want to point out that the rebuttal and discussion with Reviewer WBy6 could be improved.
The rebuttal does not appear suitable for leading a constructive discussion.
The ICLR discussion process is interactive.
I would recommend using discussion to understand the reviewer's perspective and make a meaningful revision, rather than disagreeing or pointing out the review's flaws.

The paper exhibits strong novelty and extensibility, but also suffers from insufficient evaluation and analysis as weaknesses.
It is close to borderline, but the overall rating leans towards acceptance rather than rejection.
Thus, I recommend acceptance (poster)

**Reviewer Concerns:**

Resolved concerns

- Reviewer BQcd
  - Presentation needs to be improved: unclear $D_p$ in 4.2 and concept figure
  - OOD reasoning benchmark in 4.2
  - ViT-L experiments in 4.4

Remaining concerns

- Reviewer BQcd
  - Comparisons with baselines are not sufficient
- Reviewer WBy6
  - Not enough quantitative evaluation
  - Justification for cosine similarity
  - Sensitivity analysis against K and s
  - Unclear definition of "concept-agnostic"

**Reviewer Scores:**

- Reviewer BQcd: Would raise the score. The rebuttal provides additional experiments and explanations. Although the new baseline is not added to the comparison, the rebuttal offers a reasonable explanation.
- Reviewer HhFz: Would maintain the current score. The reviewer raised only minor concerns, and the rating is positive.
- Reviewer WBy6: Would maintain the score. The rebuttal failed to address the reviewer's concern. The reviewer's comments clarify that the major concerns remain unaddressed.

---

### Decision · Program_Chairs · 2026-01-26

Accept (Poster)